# SLUCM+BEM (v1.0): A simple parameterisation for dynamic anthropogenic heat and electricity consumption in WRF-Urban (v4.3.2)

Yuya Takane[1], Yukihiro Kikegawa[2], Ko Nakajima[1], Hiroyuki Kusaka[3]

[1]Environmental Management Research Institute, National Institute of Advanced Industrial Science and Technology (AIST), Tsukuba, 305-8569, Japan
[2]School of Science and Engineering, Meisei University, Tokyo, 191-8506, Japan
[3]Center for Computational Sciences, University of Tsukuba, Tsukuba, 305-8577, Japan

*Correspondence to*: Yuya Takane (takane.yuya@aist.go.jp)

**Abstract.** We propose a simple dynamic anthropogenic heat ($Q_F$) parameterisation for the Weather Research and Forecasting (WRF)-single-layer urban canopy model (SLUCM). The SLUCM is a remarkable physically based urban canopy model that is widely used. However, a limitation of SLUCM is that it considers a statistically based diurnal pattern of $Q_F$. Consequently, $Q_F$ is not affected by outdoor temperature changes and the diurnal pattern of $Q_F$ is constant throughout the simulation period. To address these limitations, based on the concept of a building energy model (BEM), which has been officially introduced in WRF, we propose a parameterisation to dynamically and simply simulate $Q_F$ from buildings ($Q_{FB}$) through physically based calculation of the indoor heat load and input parameters for BEM and SLUCM. This method allows users to simulate the dynamic $Q_F$ and the electricity consumption ($EC$) as the outdoor temperature, building insulation, and heating and air conditioning (HAC) performance change. This is achieved via simple selection of certain $Q_F$ options among the urban parameters of WRF. SLUCM+BEM was shown to simulate temporal variations of $Q_{FB}$ and EC for HAC ($EC_{HAC}$) and broadly reproduce the $EC_{HAC}$ estimates of more sophisticated BEM and $EC_{HAC}$ observations in the world's largest metropolis, Tokyo.

## 1 Introduction

In the current era of climate change, cities are among the most critical sites for climate change mitigation and adaptation. With urban development, population concentration and urban warming, cities consume more energy and emit more greenhouse gases (GHGs) and anthropogenic waste heat ($Q_F$) than ever. As a result, global and local urban warming will continue to increase (IPCC, 2021; Takane et al., 2019; 2020; Kikegawa et al., 2022). Against this backdrop, climate change mitigation efforts toward the goal of carbon neutrality by 2050 are gaining momentum in countries across development stages, and urban climate change adaptation efforts are also progressing. However, in countries and regions where urban areas are expanding due to population and economic growth, GHG and $Q_F$ emissions associated with urbanisation are expected to continue to increase. In addition, energy consumption, particularly for air conditioning (AC), is predicted to increase under continued global warming in developed and other countries (IEA 2018). Therefore, clarifying the current state of energy consumption,

climate, and GHG emissions in urban areas and projecting these factors into the future are essential strategies toward climate change mitigation and adaptation, particularly for the development of a global climate change mitigation plan to achieve carbon neutrality by 2050.

Urban canopy models (UCMs) represent a valuable method for physically estimating and projecting urban warming, urban heat islands (UHI), and energy consumption (e.g., Kusaka et al., 2001; Chen et al., 2011). The UCM is an essential physical parameterisation for the calculation of urban weather and climate, including the UHI effect. Several UCMs have been developed by researchers worldwide and intercomparison experiments have been conducted (Grimmond et al., 2010; 2011; Lipson et al., 2023). Among these models, some UCMs have been officially implemented in the Weather Research and Forecasting (WRF) model (Skamarock et al., 2021) and have many users worldwide (Chen et al., 2011). WRF employs two main UCM options: the UCM alone, and a combined building energy model (BEM). The UCM alone corresponds to the single-layer UCM (SLUCM, Kusaka et al., 2001; Kusaka and Kimura, 2004), and a building effect parameterisation (BEP) (Martilli et al., 2002), whereas in the combined building energy model, the BEM is coupled to the BEP to construct BEP+BEM (Salamanca et al., 2010). Both UCM options have advantages and disadvantages.

The advantages of the SLUCM are that it requires fewer input parameters and has lower computational cost than the combined building energy model. However, in SLUCM, $Q_F$ adopts a user-set diurnal pattern (Table 1). Thus, $Q_F$ does not follow outdoor temperature changes, and the diurnal pattern of $Q_F$ is constant throughout the simulation period.

By contrast, the advantages of the BEP+BEM model are that the heat emitted by buildings ($Q_F$ from buildings [$Q_{FB}$]) varies with the outdoor temperature and human activity, allowing for dynamic calculation; and that electricity consumption ($EC$) associated with heating and AC (HAC) (i.e., $EC_{HAC}$) can be calculated (Table 1). However, the limitations of BEP+BEM are that $Q_F$ from traffic is not considered, the BEM has numerous input parameters, and obtaining realistic parameter settings is difficult. Although calculations can be performed with default parameter inputs, the results of such calculations significantly overestimate measured $EC$ when default parameters are entered (e.g., Takane et al., 2017; Xu et al., 2018). One suggested cause of this overestimation is that the setting (assuming an unrealistic situation) is based on the constant use of AC on all floors and in all buildings (Takane et al., 2017; Xu et al., 2018).

The aim of this study was to propose a new parameterisation, SLUCM+BEM, which exploits the advantages of both SLUCM and BEP+BEM, while compensating for the shortcomings of both models.


**Table 1: Description of urban canopy parameterisations.**

| | SLUCM[1] | **SLUCM+BEM** | BEP+BEM[2] | CM-BEM[3] | CLMU[4, 5] | BEM-TEB[6] |
|---|---|---|---|---|---|---|
| $Q_F$ from buildings | Prescribed | **Dynamic** | Dynamic | Dynamic | Dynamic | Dynamic |
| $Q_F$ from traffic | Prescribed | **Prescribed** | – | Prescribed | Prescribed | Prescribed |
| Internal heat gain | – | **Input** | Input | Input | – | Input |
| $EC_{HAC}$ | – | **Dynamic** | Dynamic | Dynamic | Dynamic | Dynamic |
| Partial AC | – | **Implemented** | – | Implemented | Implemented | – |
| COP | – | **Dynamic** | Constant | Dynamic | Constant | Dynamic |
| Cooling tower | – | **Implemented** | – | Implemented | – | – |
| Windows | – | – | Implemented | Implemented | – | Implemented |
| Ventilation | – | – | Implemented | Implemented | Implemented | Implemented |
| Weekday–weekend difference | – | – | – | Implemented | – | – |

AC, air conditioning; BEM, building energy model, BEP, building effect parameterisation; CLMU, community land model–urban; CM, canopy model; COP, coefficient of performance; EC, electricity consumption; $Q_F$, anthropogenic heat, SLUCM, single-layer urban canopy model; TEB, town energy balance.

[1] Kusaka et al. (2001), [2] Salamanca et al. (2010), [3] Kikegawa et al. (2003), [4] Oleson and Feddema (2020), [5] Li et al. (2024), [6] Bueno et al. (2012).

The SLUCM+BEM proposed in this study has two main characteristics (Table 1). First, it resolves a limitation of SLUCM, the user-defined diurnal pattern of $Q_F$ during the simulation/prediction period. Specifically, by introducing the BEM concept (Kikegawa et al., 2003; 2006; Salamance et al., 2010; Bueno et al., 2012; Oleson and Feddema, 2020), heat conduction through the wall and roof is calculated from the difference between the outdoor air temperature and the building boundary temperature in the urban canopy space, and this value and the indoor heat load are processed by HAC to calculate $EC_{HAC}$, thereby enabling dynamic calculation of $EC$ and $Q_{FB}$. As a result, improved accuracy can be expected on days that deviate from the average conditions during the simulation period, such as hot or cold days.

Second, SLUCM+BEM considers partial AC (in which AC is not used at all times, on all floors, or in all buildings), coefficient of performance (COP) changes and cooling towers, similar to CM-BEM (Kikegawa et al., 2003; Takane et al., 2022; Nakajima et al., 2023), which is among the most detailed urban models incorporating a canopy model (CM) and BEM in use today. Nevertheless, the parameterisation has been kept as simple as possible, e.g., by not considering windows, which require

uncertain parameter inputs. In this manner, the advantages of BEP+BEM described above were exploited, and the corresponding disadvantages were overcome.

As shown in Table 1, the SLUCM+BEM proposed in this study has similar characteristics to CM-BEM. However, SLUCM+BEM is simpler than CM-BEM. A typical simplification is the absence of windows in the buildings (such that the amount of solar radiation entering the building is not considered in the calculation of the indoor heat load). Although a previous

study improved the SLUCM and introduced a detailed window sub-model in their BEM-SLUCM, which is used only for offline simulations (Chen et al., 2021), it should be noted that many offices and homes use window coverings during summer, and that incoming solar radiation becomes small during winter. Moreover, this assumption has been used in many similar models such as the community land model–urban (CLMU; Oleson et al., 2008, Oleson and Feddema, 2020, Li et al. 2024) and urban climate and energy model (UCLEM; Lipson et al., 2018). Furthermore, SLUCM+BEM is intended to be used in cities

worldwide and a database of global window areas does not yet exist. Therefore, these parameters cannot be set properly, which may lead to results with large uncertainties. This shortcoming is unavoidable and reasonable at present, as SLUCM+BEM is intended for use in cities worldwide.

During the development of SLUCM+BEM, emphasis was placed on minimising the number of new parameters to be entered and simplifying its use compared to the original SLUCM and BEP+BEM models, as well as on careful comparison of

SLUCM+BEM with the CM-BEM and observed data. Specifically, we sought to render SLUCM+BEM usable by those who employ both WRF and the original SLUCM. Users simply change certain $Q_F$ options (AHOPTION) in the urban parameter setting file (URBPRAM.TBL) of WRF 1 and 2 (please see Section 2.1).

There is significant importance in updating SLUCM, which has users worldwide, e.g., in Europe (Loridan et al., 2010; Tsiringakis et al., 2019), Asia (Miao et al., 2009; Takane and Kusaka, 2011; Kusaka et al., 2012; 2014; Adachi et al., 2014;

Doan et al., 2019), North America (Georgescu et al., 2014; Krayenhoff et al., 2018), Oceania (Hirsch et al., 2021), and South America (Umezaki et al., 2020) and is preferred by more than 90% of its users (NCAR, 2015). A recent systematic review reported that WRF coupled with SLUCM is the most commonly applied numerical tool for urban environmental studies at the city and regional scales (Krayenhoff et al., 2021). In particular, the development of SLUCM+BEM will improve the applicability of the WRF model by supporting the prediction and estimation of $EC$ and $Q_{FB}$ emissions and will also drive shifts

in the consumer sector toward carbon neutrality. Furthermore, this improvement will be applicable not only to the Tokyo metropolitan area, which is the target of this study, but to cities worldwide.

Notably, $Q_{FB}$ and $EC$ calculated in SLUCM+BEM are based on HAC use, which seems appropriate given the rapid spread of HAC driven by climate change and economic growth, and the background that heat pumps are positioned as renewable energy in the European Union and are widely used for heating. The same assumption is used in BEP+BEM.

## 2 Methods

### 2.1 Model development

An overview of SLUCM+BEM is provided in Fig. 1. In conventional SLUCM, users turn the consideration of sensible $Q_F$ off or on by selecting 0 or 1 as the AHOPTION option in the URBPRAM.TBL setting, respectively. For AHOPTION = 1, hourly values of sensible $Q_F$, given as the product of its daily maximum (AH) and hourly variation factor (AHDIUPRF), which are both prescribed in URBPRAM.TBL, are added to the sensible heat flux $Q_H$ calculated by SLUCM, thereby returning $Q_F$ to the atmospheric first layer of the WRF (Fig. 1a). Users also set the building indoor boundary conditions BOUNDR for roofs and BOUNDNB for walls (hereafter referred to collectively as BOUND*) to 1 or 2, referred to in Fig. 1 as "zero-flux" and "constant", respectively. The default setting is BOUND* = 1 (i.e., zero-flux).

With BOUND* = 1 (i.e., zero-flux; Fig. 1a), the conductive heat fluxes through walls and roofs at indoor boundaries are zero due to equilibrium between the indoor boundary temperature (K) (TBLEND for walls and TRLEND for roofs) and the temperature (K) at the fourth layer of walls and roofs (TBL(4) and TRL(4), respectively). Therefore, the simulation assumes perfect insulation performance under this setting. With BOUND* = 2 (constant; Fig. 1b), the values of TBLEND are constant, allowing for imbalance with TBL(4) and thus generating conductive heat fluxes at indoor boundaries. If the outdoor temperature in the urban canopy space is higher than the value of TBLEND set in URBPRAM.TBL (often in daytime during summer), conductive heat flux can penetrate indoors and then disappear from the model, making buildings behave as heat sinks (i.e., the user-set $Q_F$ assumes that such heat can contribute to $Q_F$ from air conditioners). By contrast, when the outdoor temperature is lower than the value of TBLEND (often in winter), the opposite is true: the building becomes a heat source (i.e., the building represents a heat-producing object in the urban canopy space).

At the core of the proposed SLUCM+BEM is a concept that solves the issue of energy imbalance described above and obtains a more realistic energy budget for buildings under the conditions of HAC by estimating the amount of heat sink or source that the buildings provide under the conventional SLUCM setting of BOUND* = 2 (constant) and returning a part of this heat to the urban canopy space. To achieve this aim, the model calculates conductive heat fluxes through walls and roofs, estimates the indoor heat load and calculates $Q_F$ and $EC$ associated with HAC (Fig. 1c). The addition of these newly calculated variables and newly introduced parameters in SLUCM+BEM allows the model to conduct dynamic calculation of $Q_F$ and $EC$ for each time and day.

**a**

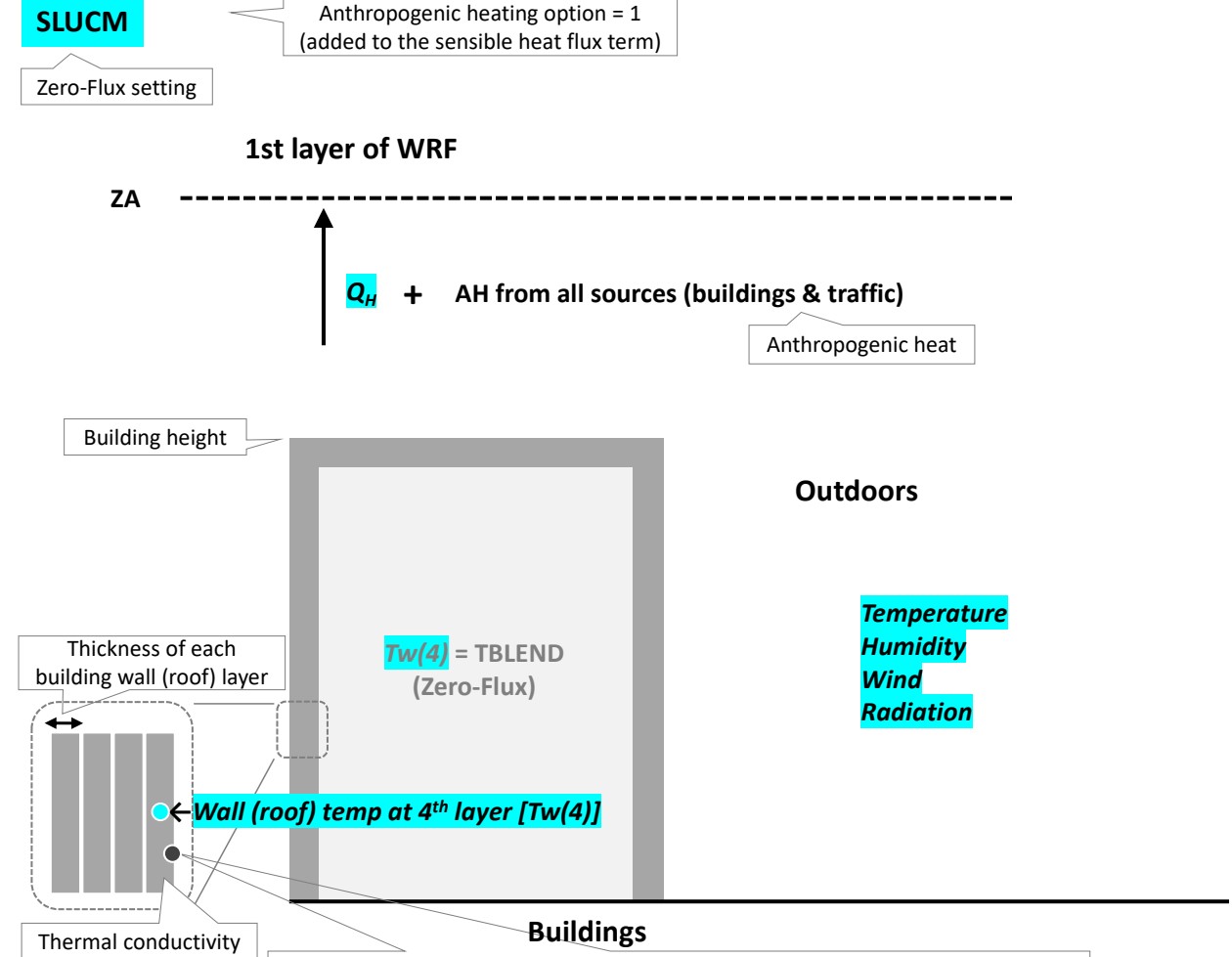

SLUCM

Anthropogenic heating option = 1
(added to the sensible heat flux term)

Zero-Flux setting

**1st layer of WRF**

ZA

$Q_H$ + **AH from all sources (buildings & traffic)**

Anthropogenic heat

Building height

**Outdoors**

Thickness of each
building wall (roof) layer

$Tw(4)$ = TBLEND
(Zero-Flux)

*Temperature*
*Humidity*
*Wind*
*Radiation*

*Wall (roof) temp at 4th layer [Tw(4)]*

Thermal conductivity
Heat capacity

**Buildings**

Lower boundary condition for building wall (roof) temperature [TBLEND (TRLEND)]

**b** SLUCM

Constant setting

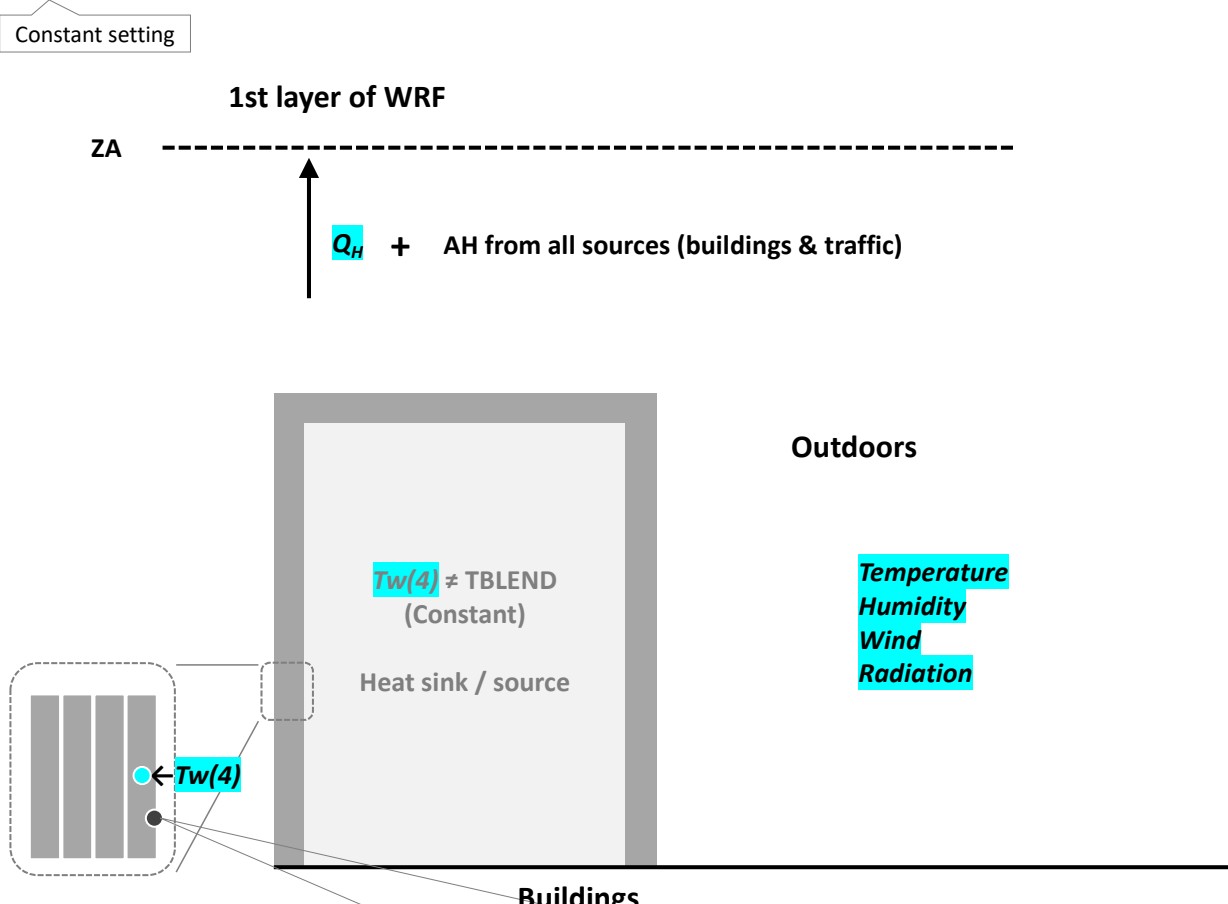

**1st layer of WRF**

ZA ----------------------------------------------

$Q_H$ **+ AH from all sources (buildings & traffic)**

**Outdoors**

*Tw(4)* ≠ TBLEND
(Constant)

Heat sink / source

*Temperature*
*Humidity*
*Wind*
*Radiation*

←*Tw(4)*

**Buildings**

Lower boundary condition for building wall (roof) temperature [TBLEND (TRLEND)]

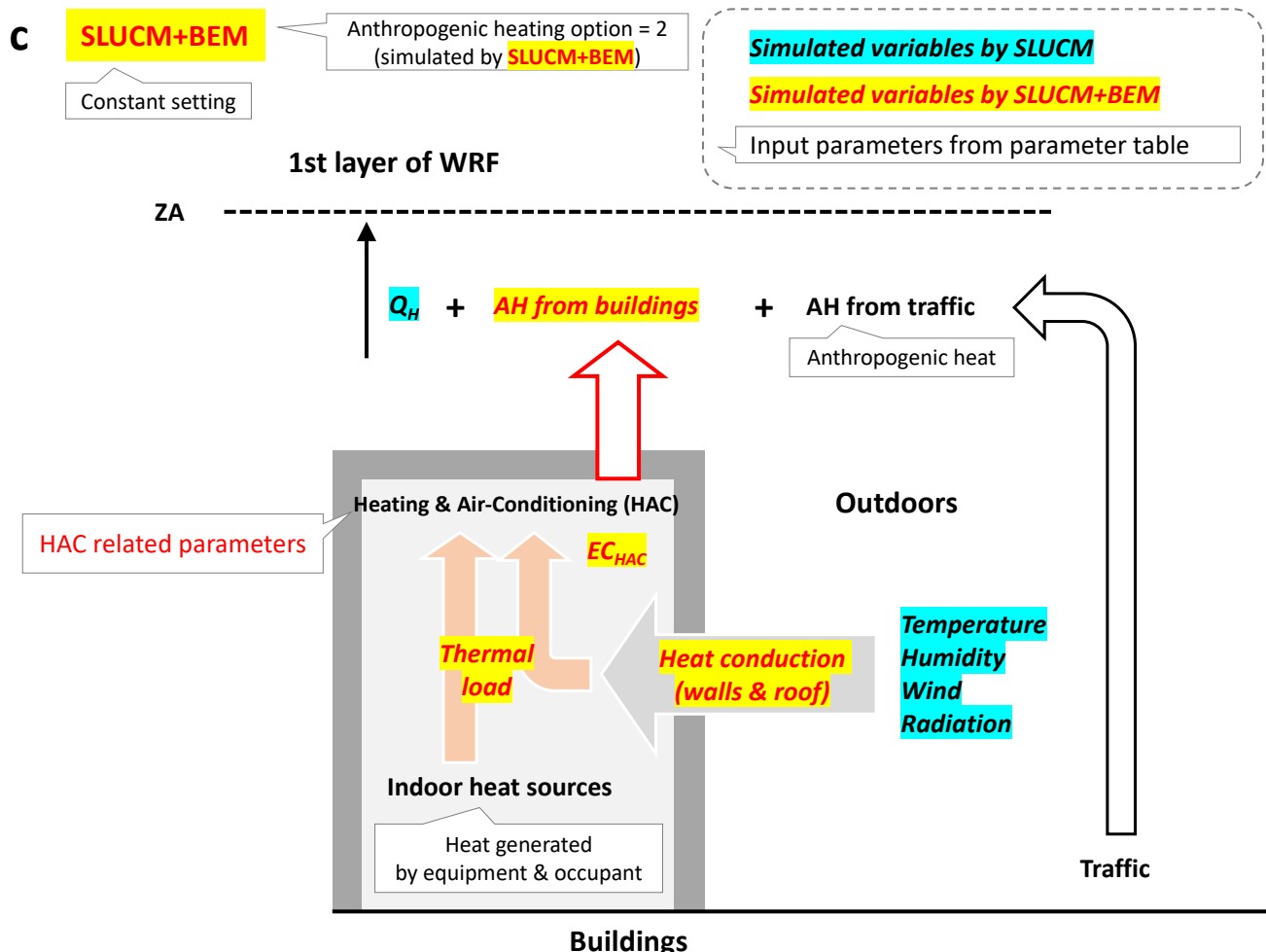

**Figure 1: Schematic of energy budgets for an urban canopy layer that includes buildings. The single-layer urban canopy model (SLUCM) with (a) "Zero-Flux" and (b) "Constant" settings. (c): The updated SLUCM based on a building energy model (BEM), thus SLUCM+BEM, with a "Constant" setting. Blue and yellow highlighting indicate variables simulated by SLUCM and SLCUM+BEM respectively. The text in the callouts indicates original or newly introduced inputs to the WRF parameter table URBNPRAM.TBL.**

Conductive heat transfer (*HTRANS*) is estimated as follows:

$$HTRANS = 2h\ AKSB\left(\frac{TBL(4) - TBLEND}{(\frac{DZB(4)}{2})}\right) + r\ AKSR\left(\frac{TRL(4) - TRLEND}{(\frac{DZR(4)}{2})}\right) \tag{1}$$

where the first and second terms on the right-hand side are conductive heat fluxes through walls and roofs, respectively; $h$ and $r$ are the normalised building height and roof width, respectively, as defined by Kusaka et al. (2001); $AKSB$ and $AKSR$ are the thermal conductivity of walls and roofs (W m$^{-1}$ K$^{-1}$), corresponding to $\lambda_W$ and $\lambda_R$ in Kusaka et al. (2001), respectively; and $DZB$ and $DZR$ are the thickness of each layer of walls and roofs, respectively.

Following the estimation of $HTRANS$, indoor sensible heat load ($H_{in}$; positive in summer and negative in winter) is calculated as follows:

$$H_{in} = HTRANS + A_f qE + A_f P \varphi_P q_{hs}$$

(2)

where the right-hand side shows each component of indoor sensible heat load. The first term is the $HTRANS$ estimated using Eq. (1). The second and third terms are internal sensible heats generation by the equipment and the occupants respectively (always positive). In the terms, $A_f$ is the floor area (m$^2$); $qE$ is the sensible heat gain from appliances per floor area (W m$^{-2}$); $P$ is the peak number of occupants per floor area (person m$^{-2}$); $\varphi_P$ is the ratio of hourly occupants to $P$ (dimensionless); and $q_{hs}$ is the sensible heat generation from building occupants (W person$^{-1}$). For simplification, the model does not consider the transmission of solar insolation through windows or sensible heat exchange through ventilation.

Previous studies have reported that because BEP+BEM assumes central, rather than decentralised, HAC systems, BEP+BEM cannot distinguish between rooms with and without individual HAC units, leading to overestimations of $EC_{HAC}$ (Takane et al., 2017; Xu et al., 2018). Accordingly, HAC systems are assumed to operate in all buildings, floors, and rooms in BEP+BEM. This situation is not common in Asian cities, where mainly individual HAC units are used (e.g., Ihara et al., 2008; Kikegawa et al., 2014). Thus, to prevent overestimation of HAC use and improve the reproducibility of $EC_{HAC}$, we introduced the following three parameters, as described by Takane et al. (2017), considering the use of decentralised HAC systems: the ratio of abandoned houses/buildings to all houses/buildings (parameter a, AB_BUILD_RATIO), the ratio of air-conditioned floor area to total floor area (parameter b, AC_FLOOR_RATIO), and the ratio of electric HAC usage for cooling or heating to all cooling or heating equipment (parameter c, AC_USAGE_RATIO_CL and AC_USAGE_RATIO_HT for cooling and heating, respectively). Settings for these parameters are provided in Table 2. Regarding parameter a, many abandoned houses are present in Japan, which represents a social problem for the country. According to Osaka City (2015), the proportion of abandoned houses among the city's housing stock is 0.172, and it is reasonable to assume that these houses do not use HAC. For parameter b, the ratio of air-conditioned floor area to total floor area was reported by Kikegawa et al. (2014), with values of 0.71 and 0.05 in office and residential areas, respectively. Salamanca et al. (2013) also considered this ratio and demonstrated that BEP+BEM could reproduce the diurnal profile of electricity demand for AC when the value was set to 0.65 for the city of Phoenix, Arizona, USA. Regarding parameter c, most people use electric AC as cooling equipment during summer, whereas few people use electric AC systems as heat pumps during winter, as many other types of heating equipment

are available. We used parameters a, b, and c to calculate the sensible heat load processed by HAC systems ($H_{out}$; positive in summer, negative in winter) as follows:

$$H_{out} = H_{in} \times (1 - a) \times b \times c. \tag{3}$$

We calculated $EC$ for HAC ($EC_{HAC}$) as follows:

$$EC_{HAC} = \frac{|H_{out}|}{COP}. \tag{4}$$

The coefficient of performance ($COP$) of the HAC system in Eq. (4) is realistically reproduced by the following equation, after Kikegawa et al. (2005):

$$COP = \frac{rCOP \times fq \times z}{fp \times fx}, \tag{5}$$

where $rCOP$ is the nominal COP of the considered HAC system; $fq$ and $fp$ respectively represent the dependency of the heating or cooling capacity and $EC$ of the system on its operational conditions as functions of the dry-bulb outdoor air temperature and the wet-bulb indoor air temperature; $z$ is the part-load ratio of the system; and $fx$ represents the dependency of $fp$ on $z$. The functions $fq$, $fp$, and $fx$ were taken from Kikegawa et al. (2005) for typical Japanese HAC systems, as was $rCOP$.

Using $H_{out}$ (Eq. 3), $EC_{HAC}$ (Eq. 4), and $COP$ (Eq. 5), the anthropogenic heat ($Q_F$) from buildings ($Q_{FB}$; positive in summer, negative in winter) was calculated at each time step as follows:

$$Q_{FB} = H_{out} + EC_{HAC} = \frac{COP+1}{COP} H_{out} \quad \text{; during cooling operation (summer)} \tag{6}$$

$$Q_{FB} = H_{out} - EC_{HAC} = \frac{COP-1}{COP} H_{out} \quad \text{; during heating operation (winter)} \tag{7}$$

In the Northern Hemisphere, this study assumes the use of cooling during June–September and the use of heating during November–March. In the Southern Hemisphere, the use of cooling is assumed for November–March and the use of heating is assumed for June–September. It is also possible to set the use of cooling and heating according to the outdoor temperature calculated using SLUCM and WRF, rather than according to the month.

In business and commercial building (BC) grids, as described by Takane et al. (2017), we divided $Q_{FB}$ for cooling into sensible heat, $Q_{FB\_S}$, and latent heat, $Q_{FB\_L}$, referring to the results of Shimoda et al. (2002) as follows, whereas all of $Q_{FB}$ for heating was treated as sensible heat:

$$Q_{FB\_S} = 0.722 Q_{FB} \tag{8}$$

$$Q_{FB\_L} = 0.278 Q_{FB}. \tag{9}$$

Shimoda et al. (2002) investigated the actual use of AC including electric and gas systems in Osaka, and reported the ratio between $Q_{FB\_S}$ and $Q_{FB\_L}$ based on an inventory approach. $Q_{FB\_S}$ and $Q_{FB\_L}$ were respectively added to the sensible and latent heat fluxes, and the results returned to the atmospheric first layer of the meteorological and climate models respectively.

Note that the $Q_{FB}$ simulated by SLUCM+BEM is the anthropogenic heat from buildings. This includes the $H_{out}$ of equations (6) and (7). This definition differs from that of the anthropogenic heat flux (AHF) datasets that are focused on non-renewable, primary energy consumption (e.g. Flanner, 2009; Varquez et al., 2021).

## 2.2 Model settings

The present study used the Advanced Research WRF (ARW) ver. 4.3.2 (Skamarock et al., 2021) and online coupling of WRF with SLUCM+BEM. Figure 2 shows the finest model domain (d03), containing 251 grid points in the x and y directions, covering the Tokyo Metropolitan Area (TMA), which was the focus of our study. Domains 1 (d01) and 2 (d02) cover all of Japan and the central area of Japan, respectively. We set the horizontal grid spacing to 25, 5, and 1 km for domains d01, d02 and d03, respectively. The model top was 50 hPa, with 37 vertical sigma levels. In this simulation, the initial and boundary conditions were derived from the National Centres for Environmental Prediction Global Tropospheric Final Analysis (NCEP–FNL) from the Global Data Assimilation System with 0.25° horizontal grid spacing (GDAS, 2015), and Group for High-Resolution Sea Surface Temperature (GHRSST) Level 4 data with 1-km horizontal grid spacing (Chao et al., 2009).

The following schemes were used in the simulation: updated Rapid Radiation Transfer Model (RRTMG) short- and long-wave radiation schemes (Iacono et al., 2008), Morrison 2-moment cloud microphysics scheme (Morrison et al., 2009), Mellor–Yamada–Janjic atmospheric boundary-layer scheme (Mellor & Yamada, 1982; Janjic, 1994; 2002), Noah land surface model (Chen & Dudhia, 2001) and SLUCM (Kusaka et al., 2001; Kusaka & Kimura, 2004) or SLUCM+BEM as proposed in this study.

As in Takane et al. (2022) and Nakajima et al. (2021; 2023), building footprint (polygon) data from a geographical information system in the TMA were used to identify urban canopy geometry. The building use and total floor area for each building in the TMA were recorded in the building footprint data. Land use–land cover (LULC) datasets produced by the Geospatial Information Authority of Japan (GIAJ) (https://nlftp.mlit.go.jp/ksj/gml/datalist/KsjTmplt-L03-b-u.html, last accessed 11/09/2023) were used in this study. The urban grids were classified into three categories (C, Rm, and Rd) based on the dominant building type, as shown in Fig. 2a.

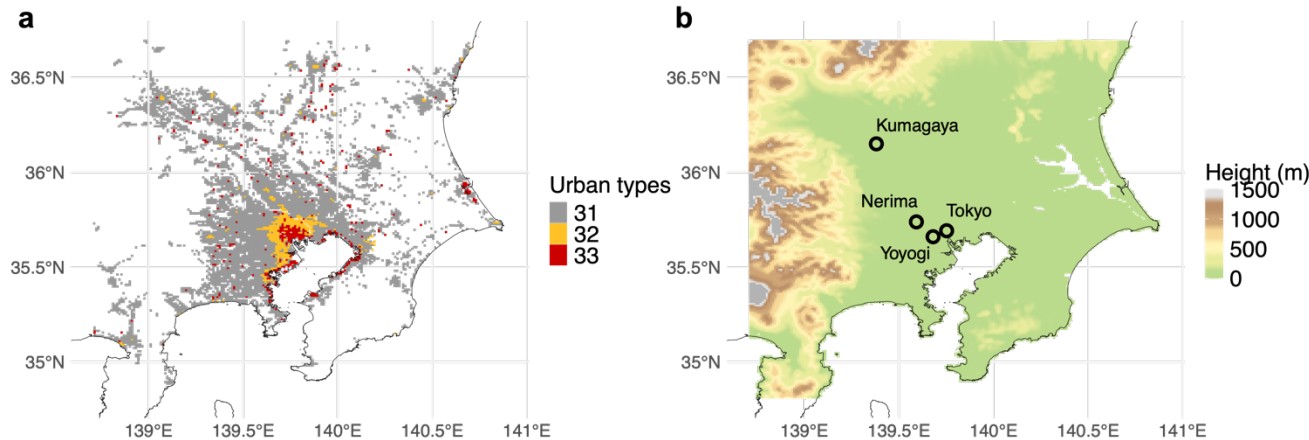

**Figure 2: Study area. (a) Distribution of three building-use categories: residential area with detached dwellings (low-density residential, 31 [grey]), residential area with multi-unit dwellings (high-density residential, 32 [yellow]), and business and commercial buildings (commercial, 33 [red]) in the Tokyo Metropolitan Area. (b) Terrain height within the study area. Open circles indicate observation sites at Nerima, Kumagaya, and Yoyogi, Tokyo.**

We also used Automated Meteorological Data Acquisition System data for TMA provided by the Japan Meteorological Agency as meteorological data for model validation.

The simulation was conducted from 09:00 JST (00:00 UTC = 09:00 JST) on 25 June to 09:00 JST on 31 August 2018 for the summer case and 25 December 2016 to 28 February 2017 for the winter case. For each case, the first 5 days were discarded as the model spin-up period. In Tokyo, the HAC is generally used only summer and winter seasons (not those of spring and autumn) (Takane et al., 2017). Spring and autumn do not affect the $EC_{HAC}$ and $Q_{FB}$ evaluations simulated by SLUCM+BEM. Thus, no 1-year simulation was performed. The 2018 and 2017 summer and winter were selected because these are the years for which the measurements of $EC$ are available (Nakajima et al., 2022), and there were more clear sky days in these than in other years.

We ran two simulation types: the original SLUCM with AHOPTION = 1 (BOUND* = 2; i.e., constant) and SLUCM+BEM with AHOPTION = 2 (BOUND* = 2; i.e., constant). The main parameters entered for each simulation type are listed in Table 2.

In the SLUCM case, $Q_F$ was an aggregate of all sources, with a maximum value (AH) and temporal variation (AHDIUPRF) for each urban category. In this study, AH and AHDIUPRF were obtained from the sum of $Q_{FB}$ calculated by CM-BEM for each grid and the separately input $Q_F$ from traffic for each building category (Nakajima et al., 2023). In the SLUCM+BEM

case, $Q_{FB}$ is the simulated variable, such that $Q_F$ from traffic was given as AH, and AHDIUPRF was the temporal pattern of $Q_F$ from traffic, in accordance with Nakajima et al., (2023). Notably, the ability to input $Q_F$ from traffic in this manner is an advantage of SLUCM+BEM over BEP+BEM (Table 1).

Both TRLEND and TBLEND are constant room temperatures, and their values are based on realistic temperature settings for HAC in Tokyo (Takane et al., 2022; Kikegawa et al., 2022; Nakajima et al. 2023). Different values were entered for summer and winter because the temperature settings of HAC systems differ seasonally.

HSEQUIP_SCALE_FACTOR and HSEQUIP are the maximum value of the internal heat gain and its percentage change over time, respectively. These parameters are used in both BEP+BEM and SLUCM+BEM without alteration. The values were obtained from actual *EC* data for the focal metropolitan area (Nakajima et al., 2023; Takane et al., 2023a).

AB_BUILD_RATIO is the ratio of abandoned houses/buildings to all houses/buildings in a city block (parameter *a* in Eq. 3). This value can be set for each urban category and was set to the value used by Takane et al. (2017).

AC_FLOOR_RATIO is the ratio of air-conditioned floor area to total floor area (parameter *b* in Eq. 3). This value can be set for each urban category and was assigned the temporally varying value for Tokyo adopted by Takane et al. (2022) and Nakajima et al. (2023).

AC_USAGE_RATIO_CL and AC_USAGE_RATIO_HT are the ratios of electric HAC use for cooling and heating to all cooling and heating equipment, respectively (parameter *c* in Eq. 3). This value can be set for each urban category and was given the value reported by Takane et al. (2017).

*rCOP* in Eq. 5 is used in BEP+BEM to indicate the performance of HAC, and SLUCM+BEM uses this parameter without alteration. Values from previous studies (Takane et al., 2017; 2023; Kikegawa et al., 2022; Nakajima et al., 2023) were employed for *rCOP*. Note that in BEP+BEM, COP is fixed at the input value of *rCOP*, whereas in SLUCM+BEM, a formula was introduced to calculate realistic COP values (Eq. 5). However, COP can also be fixed at a constant value of *rCOP* by setting COPOPTION = 0.

For both SLUCM and SLUCM+BEM, calculations are performed for two seasons, summer and winter; the TRLEND and TBLEND settings differ seasonally.


**Table 2: Parameter settings for the SLUCM and SLUCM+BEM models. The cooling and heating seasons (summer and winter) ran from 25 June to 31 August, 2018, and 25 December, 2016, to 28 February, 2017, respectively. The urban categories are: 1 low-density residential, 2 high-density residential, and 3 commercial.**

| Parameter (units) [cases] | SLUCM | | SLUCM+BEM | |
|---|---|---|---|---|
| Season | Cooling, heating | | Cooling, heating | |
| ZR (m) [Urban category = 1, 2, 3] | 7.4, 10.6, 15.2 | | | |
| FRC_URB (–) [Urban category = 1, 2, 3] | 0.7, 0.9, 0.9 | | | |
| AHOPTION (–) | 1 | | 2 | |
| AH (W m$^{-2}$) [Urban category = 1, 2, 3] | 38.8, 52.8, 141.5 in summer 19.4, 26.4, 70.7 in winter (from all sources, including buildings and traffic) | | 3.3, 7.4, 10.8 (from traffic only) | |
| AHDIUPRF (–) [Local time = hours 1–24] | 0.467 0.370 0.323 0.319 0.366 0.485 0.620 0.718 0.831 0.881 0.913 0.870 0.931 0.982 1.000 0.997 0.957 0.906 0.851 0.804 0.767 0.681 0.660 0.520 | | | |
| BOUNDR, BOUNDNB, BOUNDG (BOUND*) | 2 | | | |
| DDZR (m) [Layer = 1, 2, 3, 4] | 0.08, 0.08, 0.08, 0.08 | | | |
| DDZB (m) [Layer = 1, 2, 3, 4] | 0.06, 0.06, 0.06, 0.06 | | | |
| CAPR (J m$^{-3}$ K$^{-1}$) [Urban category = 1, 2, 3] | $0.4521 \times 10^6$, $1.588 \times 10^6$, $1.298 \times 10^6$ | | | |
| CAPB (J m$^{-3}$ K$^{-1}$) [Urban category = 1, 2, 3] | $0.674 \times 10^6$, $1.702 \times 10^6$, $1.598 \times 10^6$ | | | |
| AKSR (W m$^{-1}$ K$^{-1}$) [Urban category = 1, 2, 3] | 0.071, 0.192, 0.094 | | | |
| AKSB (W m$^{-1}$ K$^{-1}$) [Urban category = 1, 2, 3] | 0.094, 0.276 0.217, | | | |
| TRLEND (K) [Urban category = 1, 2, 3] | 300, 304, 304 for cooling | 298.15, 290.15, 290.15 for heating | 300, 304, 304 for cooling | 295815, 290.15, 290.15 for heating |
| TBLEND (K) [Urban category = 1, 2, 3] | 300, 304, 304 for cooling | 298.15, 290.15, 290.15 for heating | 300, 304, 304 for cooling | 298.15, 290.15, 290.15 for heating |
| HSEQUIP_SCALE_FACTOR | – | | 6.27, 6.84, 9.2 | |

| | | |
|---|---|---|
| (W floor-m$^{-2}$)<br>[Urban category = 1, 2, 3] | | |
| HSEQUIP (–)<br>[Local time = hours 1–24] | – | 0.76, 0.72, 0.71, 0.71, 0.72, 0.72, 0.76, 0.80, 0.86, 0.90, 0.91, 0.92, 0.91, 0.93, 0.93, 0.93, 0.96, 0.99, 1.00, 0.98, 0.94, 0.90, 0.85, 0.81 |
| AB_BUILD_RATIO (–)<br>[Urban category = 1, 2, 3] * | – | 0.136, 0.136, 0.136 |
| AC_FLOOR_RATIO (–)<br>[Urban category =1, 2, 3],<br>[Local time = hours 1–24] * | – | Urban category 1: 0.37, 0.35, 0.32, 0.31, 0.29, 0.28, 0.26, 0.24, 0.21, 0.19, 0.16, 0.16, 0.16, 0.16, 0.15, 0.15, 0.15, 0.15, 0.17, 0.18, 0.21, 0.27, 0.31, 0.34<br><br>Urban category 2: 0.41, 0.41, 0.37, 0.32, 0.30, 0.29, 0.29, 0.29, 0.29, 0.29, 0.30, 0.31, 0.31, 0.31, 0.31, 0.31, 0.31, 0.32, 0.34, 0.36, 0.38, 0.39, 0.40<br><br>Urban category 3: 0.22, 0.18, 0.17, 0.17, 0.17, 0.17, 0.17, 0.23, 0.34, 0.44, 0.51, 0.54, 0.57, 0.57, 0.57, 0.57, 0.57, 0.57, 0.57, 0.57, 0.51, 0.46, 0.40, 0.32 |
| AC_USAGE_RATIO_CL (–) [Urban category = 1, 2, 3] * | – | 1, 1, 1 |
| AC_USAGE_RATIO_HT (–) [Urban category = 1, 2, 3] * | – | 0.6, 0.6, 0.6 |
| COPOPTION (–) * | – | 1 |
| COP (–)<br>[Urban category = 1, 2, 3] | – | 5.03, 5.03, 3.58 |

AB_BUILD_RATIO, ratio of abandoned houses/buildings to all houses/buildings in a city block; AC_FLOOR_RATIO, ratio of air-conditioned floor area to total floor area; AC_USAGE_RATIO_CL, proportion of cooling AC usage; AC_USAGE_RATIO_HT, proportion of heating AC usage; AH, anthropogenic heat; AHDIUPRF, the diurnal profile of anthropogenic heating; AHOPTION, anthropogenic heating option, where 0 = no anthropogenic heating, 1 = anthropogenic heating added to the sensible heat flux term, and 2 = anthropogenic heating from buildings as simulated by SLUCM+BEM; AKSB, thermal conductivity of the building wall; AKSR, thermal

conductivity of the roof; CAPB, heat capacity of the building wall; CAPR, heat capacity of the roof; COP, coefficient of performance; COPOPTION, a switch that determines whether COP is fixed or variable, where 0 = fixed COP and 1 = COP simulated by SLUCM+BEM; DDZB, thickness of each building wall layer; DDZR, thickness of each roof layer; FRC_URB, the fraction of the urban landscape; HSEQUIP, the proportional change in HSEQUIP_SCALE_FACTOR over time; HSEQUIP_SCALE_FACTOR, peak internal heat gain; TBLEND, the lower boundary of the building wall temperature; TRLEND, the lower boundary of the roof temperature; and,

ZR, the building height.

* Newly added to SLUCM+BEM; (–) dimensionless parameter.

The SLUCM and SLUCM+BEM models were run in both offline and online modes, coupled to WRF. In offline mode, Noah-LSM (Chen & Dudhia, 2001) and SLUCM were coupled with a mosaic of natural vegetation and urban tiles, in accordance

with the online WRF land surface processes. Meteorological data measured at a flux tower in Yoyogi, Tokyo (Fig. 2b) (Hirano et al., 2015; Sugawara et al., 2021; Lipson et al., 2022) were used as forcing data in offline simulations and the results were compared with the radiation budget and heat fluxes measured at the same site. The settings for the online mode are described in Table 2. The calculated online and offline temperature and electricity consumption were compared with the corresponding measured values.

## 3 Results

### 3.1 Offline model verification

First, the offline versions of SLUCM and SLUCM+BEM were used to verify the accuracy of reproductions of the summer radiation balance and surface heat budget observed in Tokyo (Yoyogi, Fig. 2b) by Hirano et al. (2015), Sugawara et al. (2021), and Lipson et al. (2022). Their results are shown in the upper part of Fig. 3; SLUCM and SLUCM+BEM reproduced the
radiation balance and heat budgets well (Fig. 3a, b). Focusing on the sensible heat flux ($Q_H$), SLUCM somewhat overestimated the observations (Fig. 3a), whereas SLUCM+BEM reproduced them well (Fig. 3b). In addition, SLUCM was unable to calculate $EC$ (Fig. 2a), whereas SLUCM+BEM both calculated $EC$ and roughly reproduced the diurnal change of measured values in the Yoyogi area (Fig. 3b). The results of offline calculation with CM-BEM, a more sophisticated model, are shown in Fig. 3c. Both the radiation balance and surface heat budget were well reproduced, but $Q_H$ was slightly out of phase, and
SLUCM+BEM reproduced $Q_H$ better than this result; for $EC$, CM-BEM reproduced the measurements very well, whereas SLUCM+BEM showed lower accuracy. Importantly, despite the modelling simplicity of SLUCM+BEM, it captured temporal changes to some extent.

The winter results were similar to the summer results: both SLUCM and SLUCM+BEM captured features of the radiation and surface heat budgets well (Fig. 3d, e); SLUCM+BEM did not capture diurnal changes in measured $EC$, but the daily averaged
values generally aligned with observations (Fig. 3e). Notably, even the more sophisticated CM-BEM did not accurately reproduce temporal changes in winter $EC$ (Fig. 3f). Therefore, difficulty in reproducing temporal changes in winter $EC$ is not a drawback of SLUCM+BEM only.


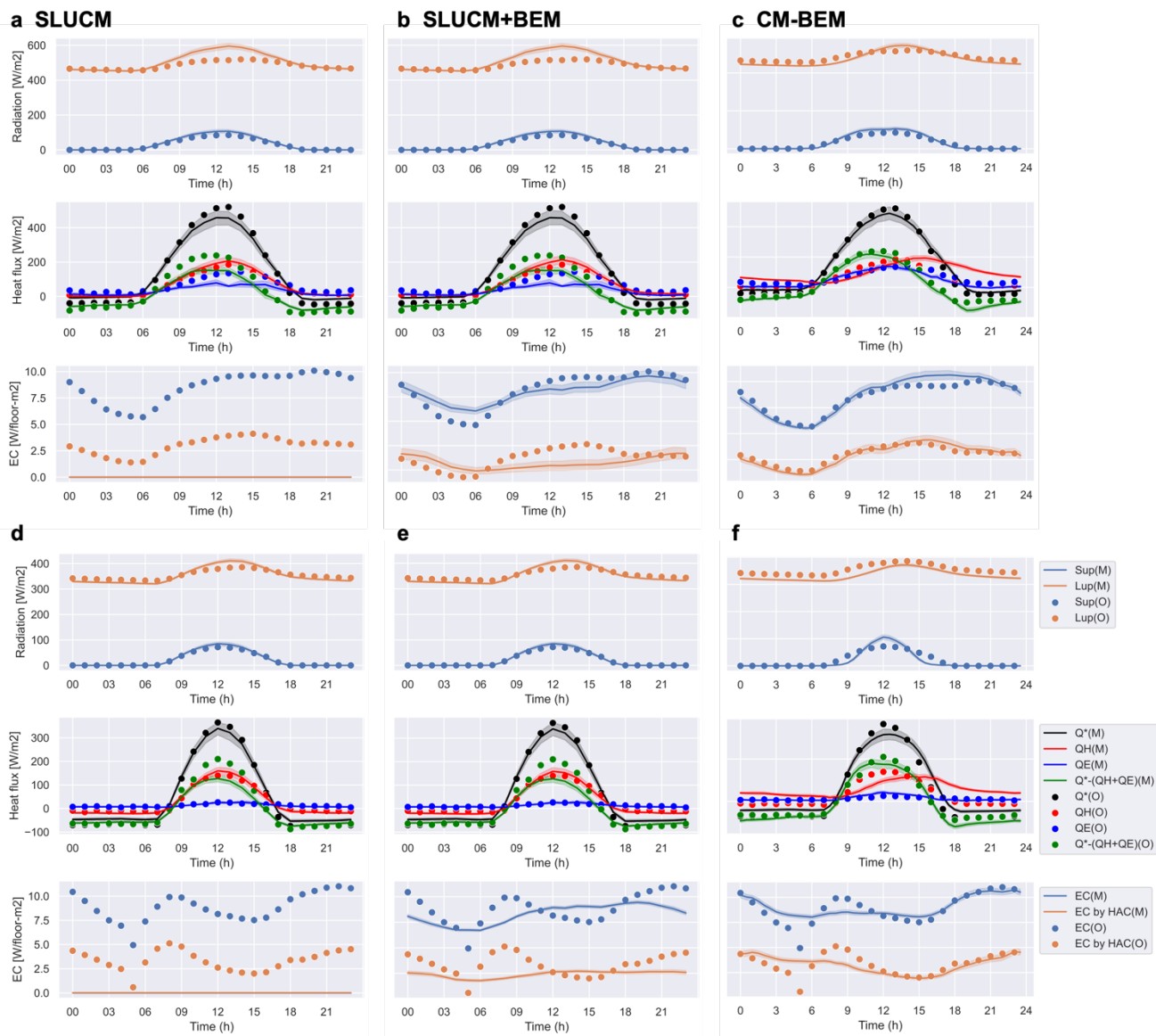

**Figure 3: Diurnal changes in radiation, surface heat balance, and electricity consumption (*EC*) in Tokyo (Yoyogi [Fig. 2b]; Sugawara et al. 2021) averaged seasonally over (a–c) summer (July–August) and (d–f) winter (January–February). Circles are observations. Lines and error bars indicate simulated average values and standard deviations from (a, d) SLUCM, (b, e) SLUCM+BEM, and (c, f) CM-BEM, respectively.**

## 3.2 Online model verification

### 3.2.1 Air temperature

This section describes the accuracy of reproducing temperatures calculated by the online model (coupled version with WRF). Figure 4a shows the temporal variation of temperature (monthly average by time of day) at three representative locations in the TMA by building use: Tokyo (BC), Kumagaya (Rm), and Nerima (Rd) (Fig. 2b), where both SLUCM (blue) and SLUCM+BEM (red) performed well in reproducing the observed temperatures (black circles), with slightly better performance by SLUCM+BEM. For example, in Tokyo, SLUCM had a mean absolute error (MAE) of 1.22°C, compared to 1.62°C for SLUCM+BEM, and little difference between the two models at the other two sites. Both models reproduced the horizontal temperature distribution in the metropolitan area better than its temporal variation. For example, SLUCM+BEM reproduced the observed urban heat island centred on Tokyo well (Fig. 5b) at 05:00 LT (when the temperature was lowest) (Fig. 5a), and observed high temperatures in the inland area at 14:00 LT (when the temperature was highest) (Fig. 5d) were similarly well reproduced (Fig. 5c).

The winter results showed a similar trend to the summer results. Both SLUCM and SLUCM+BEM captured characteristics of temporal temperature changes in Tokyo, Kumagaya and Nerima well (Fig. 4b). However, both SLUCM and SLUCM+BEM showed more significant errors for winter than for summer observations (Fig. 4a, b). The lower accuracy of winter temperature reconstructions compared to summer is not limited to SLUCM+BEM. For example, a similar trend was observed in the validation of BEP+BEM (e.g., Takane et al., 2017). Gararro & González-Cruz (2023) also reported that the introduction of electric heating reduced the peak UHI effect by 2.5–3°C. This temperature decrease during winter is due to the negative $Q_{FB}$ related to air-source heat pump AC systems used for heating. For example, the MAE of SLUCM in Tokyo was 1.69°C, whereas that of SLUCM+BEM was 2.48°C. However, this error was strongly dependent on the input parameters, such as the AH value input to SLUCM (Table 2). In general, it is not possible to precisely evaluate the success of the two models comparatively, because in summer, both models reproduced the horizontal distribution of temperature in the metropolitan area well, with SLUCM+BEM also reproducing the observed urban heat island centred on Tokyo at 05:00 and the wider temperature distribution at 14:00 (Fig. 5e–h).

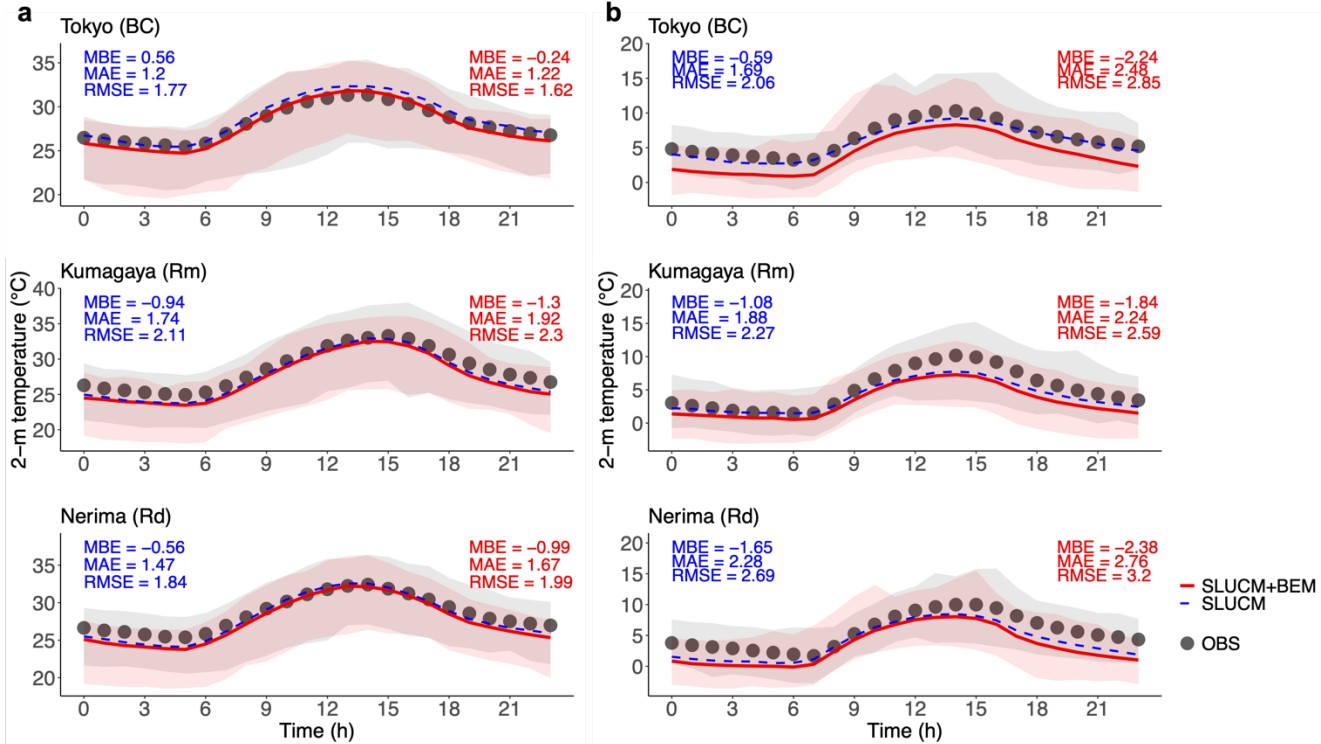

**Figure 4: Diurnal changes in 2-m temperatures in Tokyo (BC), Kumagaya (Rm), and Nerima (Rd; Fig. 2b) averaged seasonally over (a) summer and (b) winter. Circles are observations. Lines and error bars are simulated average values and 5th–95th percentiles from SLUCM (blue) and SLUCM+BEM (red), respectively. MAE, mean absolute error; MBE, mean bias error; RMSE, root mean square error.**


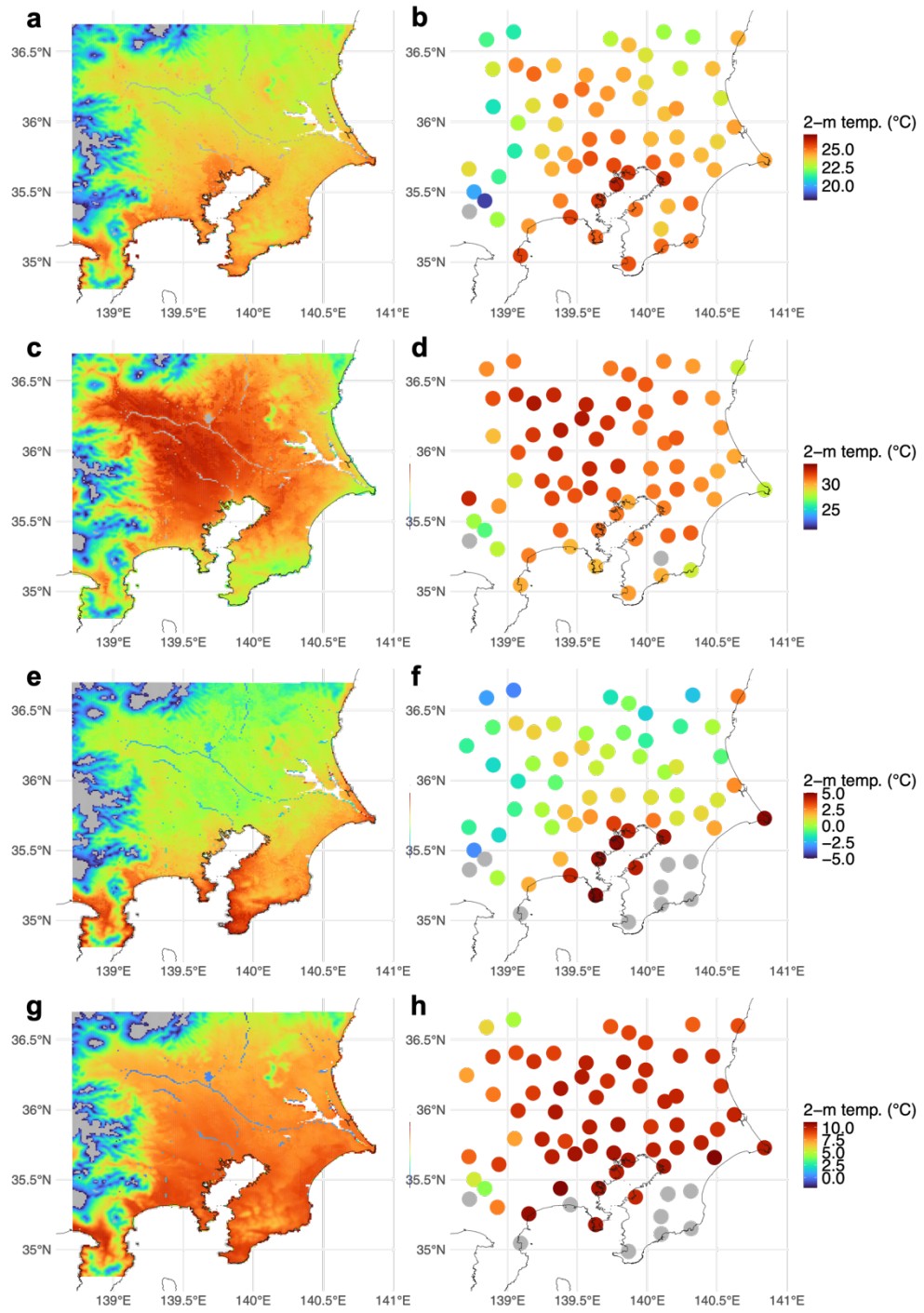

**Figure 5: Distributions of observed (right) and simulated (left) 2-m temperatures by SLUCM+BEM in the Tokyo Metropolitan Area averaged for (a, b) 05:00 local time (LT) and (c, d) 14:00 LT in summer; and (e, f) 05:00 LT and (g, h) 14:00 LT in winter.**

### 3.2.2 Electricity consumption

Notably, $EC$ cannot be calculated with the existing SLUCM. Therefore, from this point on, we report the accuracy of $EC$ reproduction only for SLUCM+BEM. In general, verifying the $Q_{FB}$ for which SLUCM+BEM performs the simulation is

difficult, because no method has been established for observing $Q_{FB}$. However, measured $EC$ data are available. In this study, high-resolution $EC$ observations for a metropolitan area reported by Nakajima et al. (2023) and Takane et al. (2023) are used to validate the accuracy of $EC$ values calculated by SLUCM+BEM. In addition, we compare the validated results of SLUCM+BEM and CM-BEM. Note that if a model can reproduce $EC$, $Q_{FB}$ can also be calculated realistically, according to Eqs. (4), (10), and (11).

We focused on validation of $EC_{HAC}$; this is the variable simulated by the models. The observed $EC_{HAC}$ was that estimated by Nakajima et al. (2022). It is better to validate $EC_{HAC}$ rather than $EC$ because $EC_{HAC}$ is the actual simulated variable; $EC$ includes input baseload parameters ("HSEQUIP_SCALE_FACTOR" and "HSEQUIP"). Thus, the $EC$ validation contains errors in both the simulated $EC_{HAC}$ and the input parameters. Nakajima et al. (2022) showed that the baseload tended to vary even among central Tokyo BC grids of the same category. CM-BEM considers baseload variability because CM-BEM inputs different

baseload values into each model grid, whereas SLUCM+BEM employs only one baseload for each urban category (the input is thus uniform across all BC grids; Table 2). Therefore, we focused only on $EC_{HAC}$ when comparing the simulated variables of SLUCM+BEM and CM-BEM. The verification focused only on the weekdays of the simulated period; the SLUCM+BEM considers only weekday conditions, as does BEP+BEM.

Figure 6a is a detailed map of the Tokyo metropolitan $EC_{HAC}$ in summer (July–August 2018 weekday average) as presented

by Nakajima et al. (2023) and Takane et al. (2023). Figure 6b is focused on central Tokyo. $EC_{HAC}$ is higher in the city centre and decreases toward the suburbs; SLUCM+BEM generally captured this (city centre > suburbs) (Fig. 6c, d vs. a, b). The $EC_{HAC}$ errors by the building type, and time, within the areas of Figure 6b and d are shown in Figure 7 (upper panel). In Rm residential grids, the daily mean bias error (MBE) was 0.8 W floor-m$^{-2}$ and the MAE 1.5 W floor-m$^{-2}$. The Rd residential grids exhibited slightly better results, with a daily MBE of –0.8 W floor-m$^{-2}$ and an MAE of 1.3 W floor-m$^{-2}$. In contrast, BC

grids yielded a daily MBE of 2.8 W floor-m$^{-2}$ and an MAE of 3.5 W floor-m$^{-2}$; the errors were greater than those of the residential grids. $EC_{HAC}$ tended to be high after 11:00 LT. Despite overestimation of the BC grids, the total, daily average errors for the areas shown in Figure 6b and d were MBE = –0.1 W floor-m$^{-2}$ and MAE = 1.5 W floor-m$^{-2}$, because the BC grid area was smaller than those of the Rm and Rd grids (Fig. 2).

The results obtained using a more detailed model, thus CM-BEM (Kikegawa et al., 2003; 2014, 2022; Takane et al., 2022;

Nakajima et al., 2023) are compared with the SLUCM+BEM data in Figure 6e and f. The CM-BEM results cover a limited area; the computational coverage is low compared to that of SLUCM+BEM. Although the areas for which $EC_{HAC}$ were calculated differ, the model resolutions (1 km) and physical parameterisations are identical, except for those of the urban

canopy and building energy models. Comparisons are possible. The CM-BEM results (Fig. 6f) well-reproduced the observations (Fig. 6b). In particular, SLUCM+BEM yielded a relatively uniform BC $EC_{HAC}$ for the city centre. In contrast, the CM-BEM values differed for each grid, in good agreement with the observations. The BC errors of CM-BEM and SLUCM+BEM were comparable; the daily MBE was 2.1 W floor-m$^{-2}$ and the MAE 2.5 W floor-m$^{-2}$. For the Rm residential grids, the daily mean errors were MBE = 0.8 W floor-m$^{-2}$ and MAE = 1.2 W floor-m$^{-2}$ (Fig. 7, bottom panel). As for the SLUCM+BEM data, the Rd residential results were slightly better than the Rm results, with daily mean errors of MBE = 0.4 W floor-m$^{-2}$ and MAE = 1.0 W floor-m$^{-2}$. As shown in Figure 6b and f, the daily average errors were MBE = 0.7 W floor-m$^{-2}$ and MAE = 1.2 W floor-m$^{-2}$, thus similar to those of SLUCM+BEM. Thus, although SLUCM+BEM is simpler than CM-BEM and can cover a larger area, it performed as well as did the detailed CM-BEM when validating $EC_{HAC}$ over the entire target area.

Note that the results presented above for CM-BEM are based on the latest version of the code, which has been improved through grid-by-grid input of internal heat gain, modelling of the AC operation schedule, and introduction of the proportion of AC systems in BC grids. Based on these improvements, the errors were reduced (Nakajima et al., 2023). These improvements provide clues for the future improvement of SLUCM+BEM.

The winter results were qualitatively similar to the summer results, but indicate somewhat better performance of CM-BEM compared to SLUCM+BEM in the simulation of $EC_{HAC}$. The distribution of winter $EC_{HAC}$ and error estimates are presented in Figs. 8 and 9, respectively.

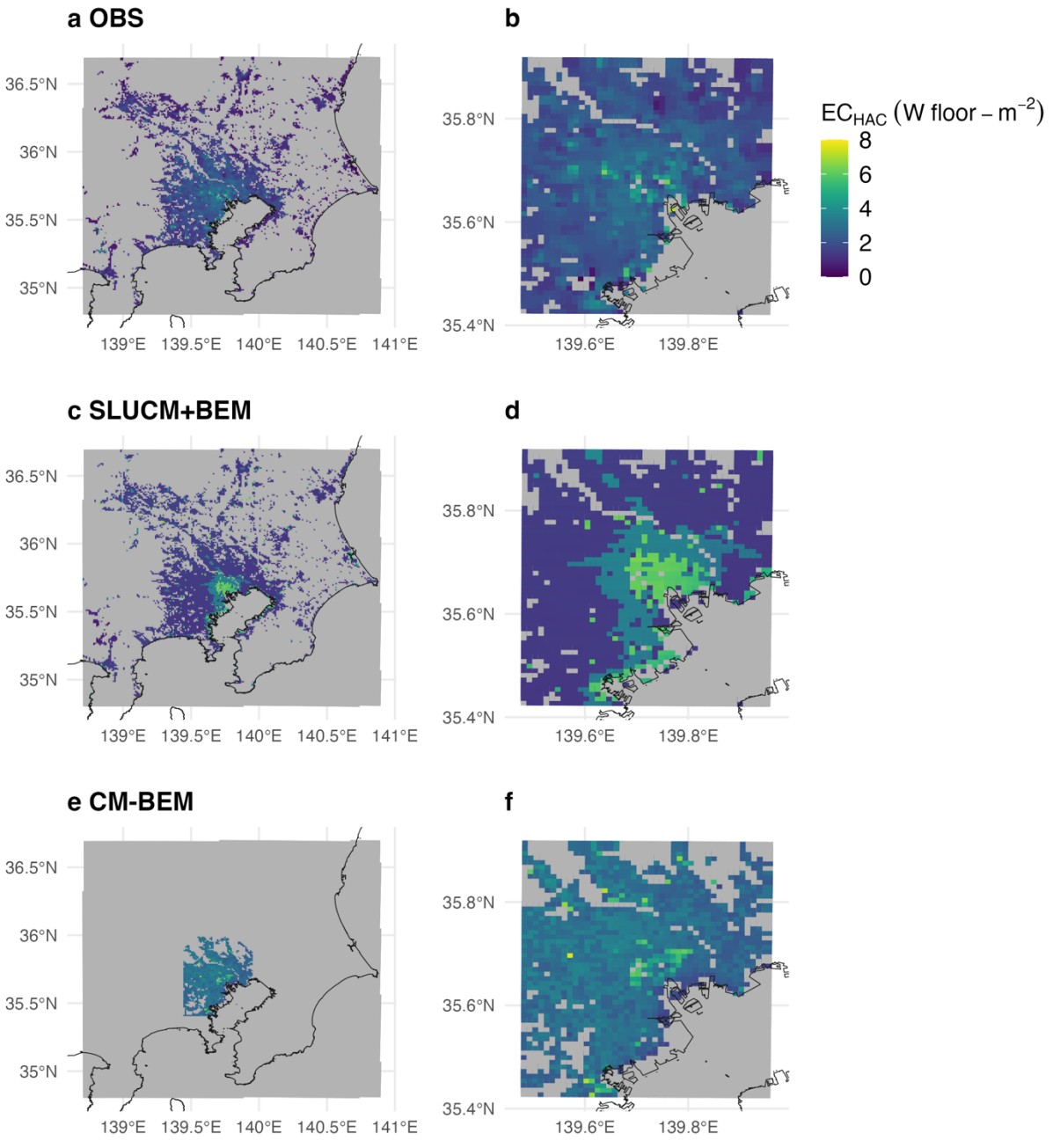

**Figure 6: Distributions of (a, b) observed and (c–h) simulated electricity consumption (*EC*) for heating and air conditioning (HAC) (i.e., *EC*ₕₐ𝒸) in the Tokyo Metropolitan Area (left) and central Tokyo area (right) averaged over the summer season's weekdays. Simulation results from (c, d) SLUCM+BEM, and (e, f) CM-BEM.**

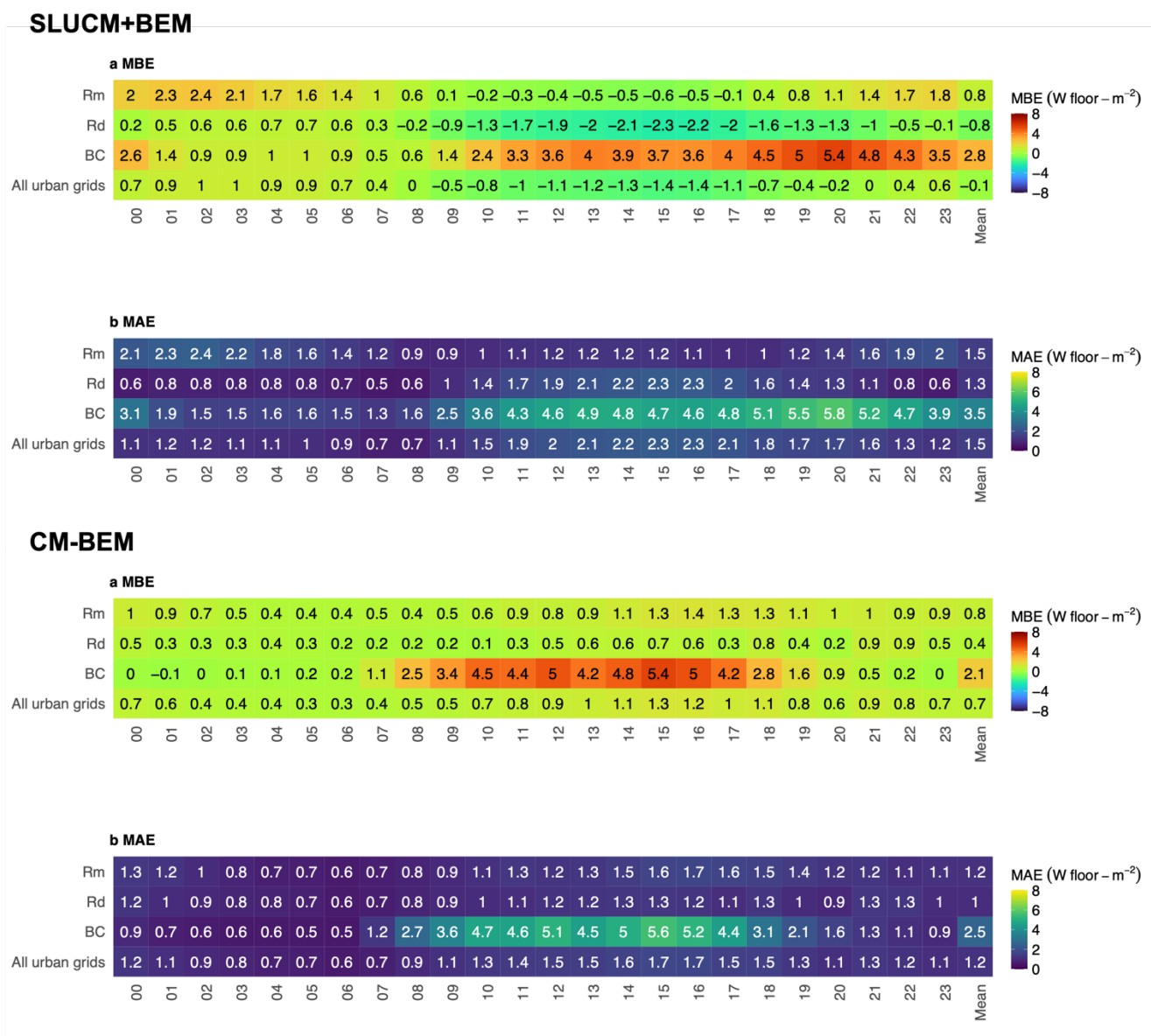

**Figure 7: Diurnal changes in (a) MBE and (b) MAE of $EC_{HAC}$ for each urban building use type, Rm, Rd, and BC, and the average of all grids from SLUCM+BEM (upper panels) and CM-BEM (new model; lower panels) averaged over the summer season's weekdays.**

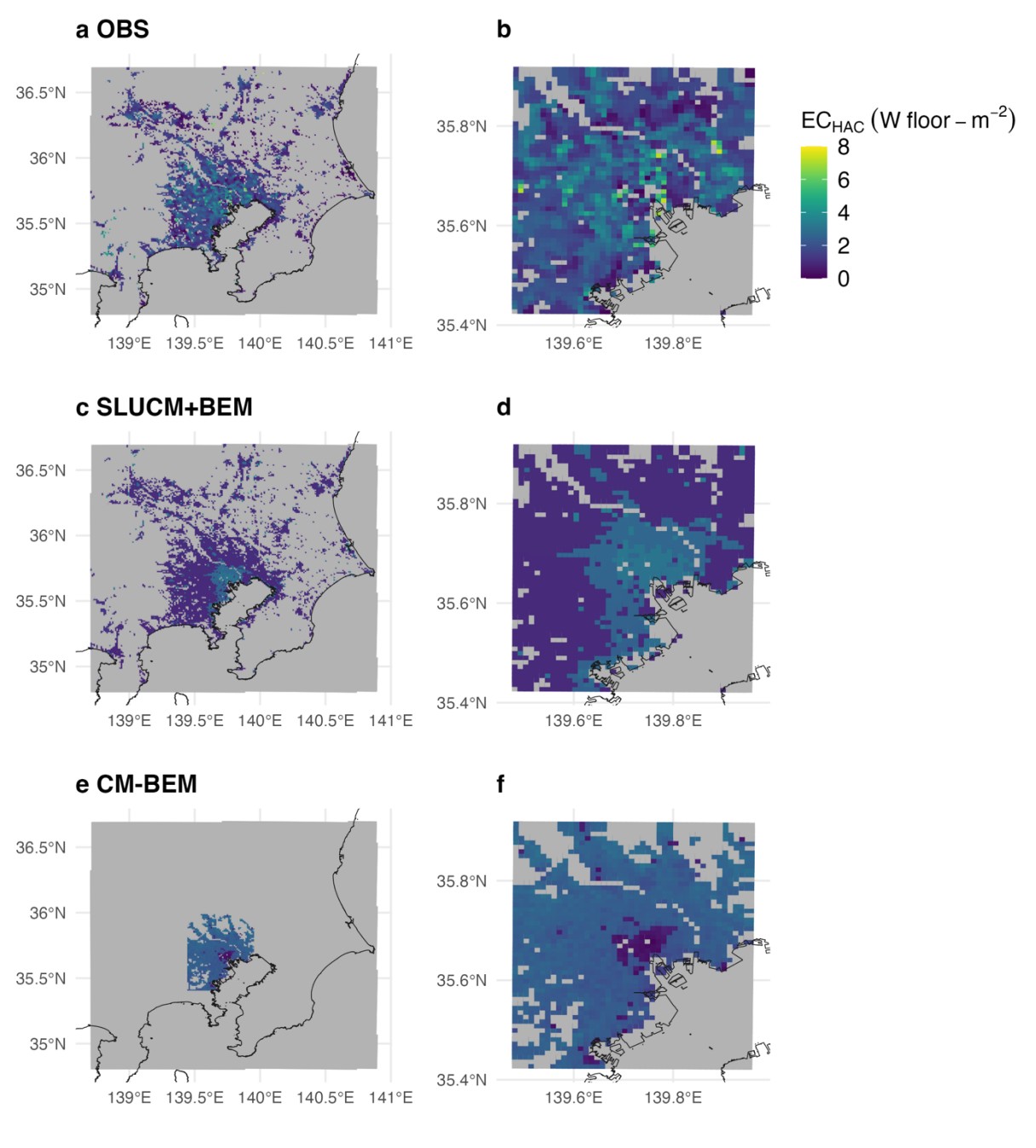

**Figure 8 As described for Fig. 6, but showing results for the winter season.**


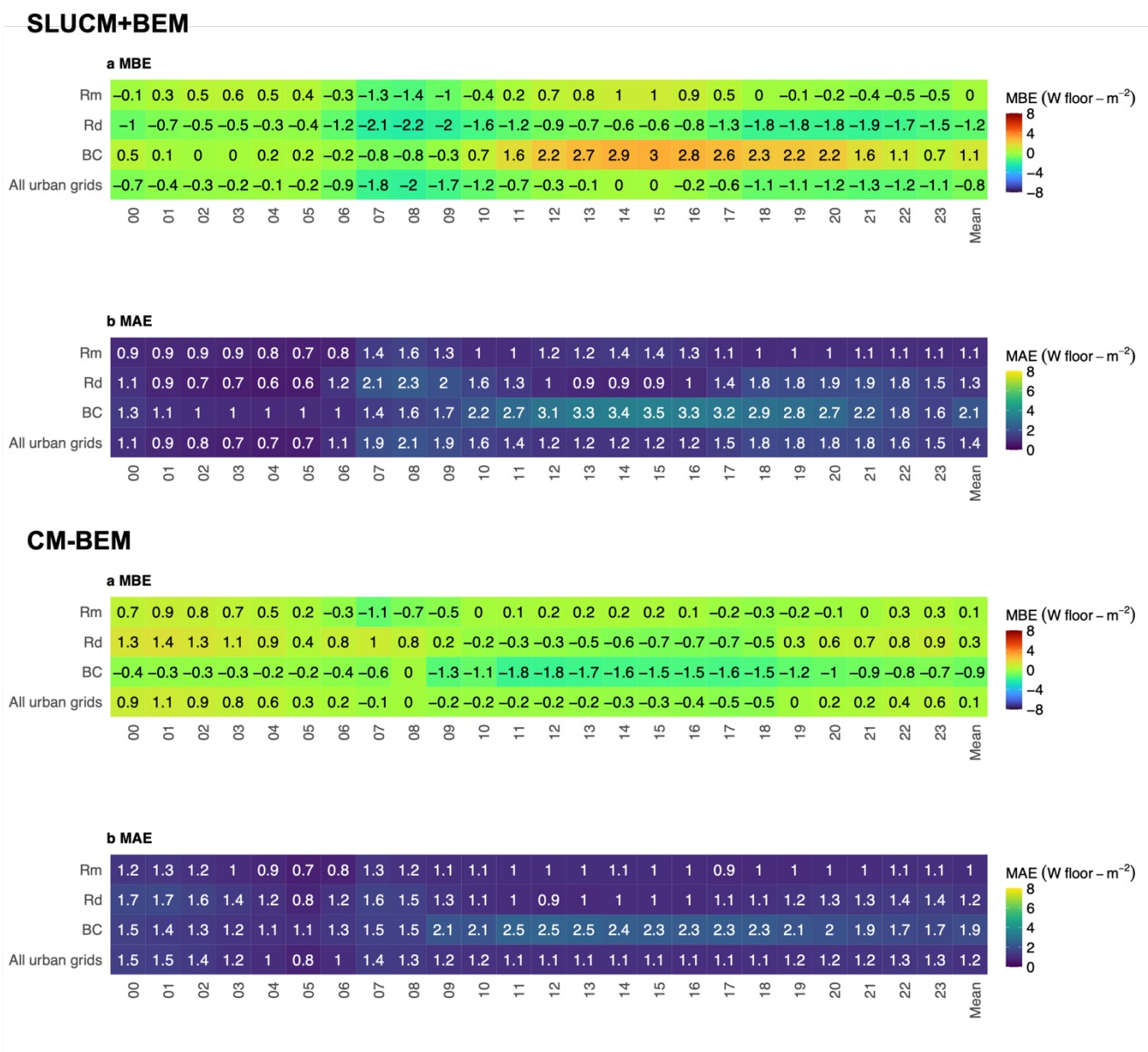

**Figure 9: As described for Fig. 7, but showing results for the winter season.**

### 3.2.2 Effects of temperature on *EC* and $Q_{FB\_S}$

The $EC_{HAC}$ calculation described above depends on the ambient temperature. The relationships between *EC* and air temperature at representative locations in Tokyo (BC), Kumagaya (Rm), and Nerima (Rd) are shown in Figure 10a. In summer, the *EC* and the temperature were positively correlated; the slope of the regression line indicates the temperature-sensitivity of *EC*

($\Delta EC/\Delta T$). Conversely, the correlation is negative in winter, and the regression line slope shallower than in summer, in part because fewer buildings use air conditioning for heating in winter than for cooling in summer (e.g. Takane et al., 2017).

The signs of the $\Delta EC/\Delta T$ values calculated by SLUCM+BEM were the same as those of the observations (positive in summer and negative in winter). The $\Delta EC/\Delta T$s simulated by SLUCM+BEM for summer are slightly overestimate in BC and Rm and underestimate in Rd, but these are reasonably good with observation (Table 3). In contract, the simulated values in winter tended to be smaller than the observations regardless urban category (Table 3). CM-BEM has the same feature as SLUCM+BEM; CM-BEM is reasonably good in summer but tended to underestimate $\Delta EC/\Delta T$ in winter. It is important to improve the $\Delta EC/\Delta T$ by SLUCM+BEM and CM-BEM especially in winter. This is a future challenge.

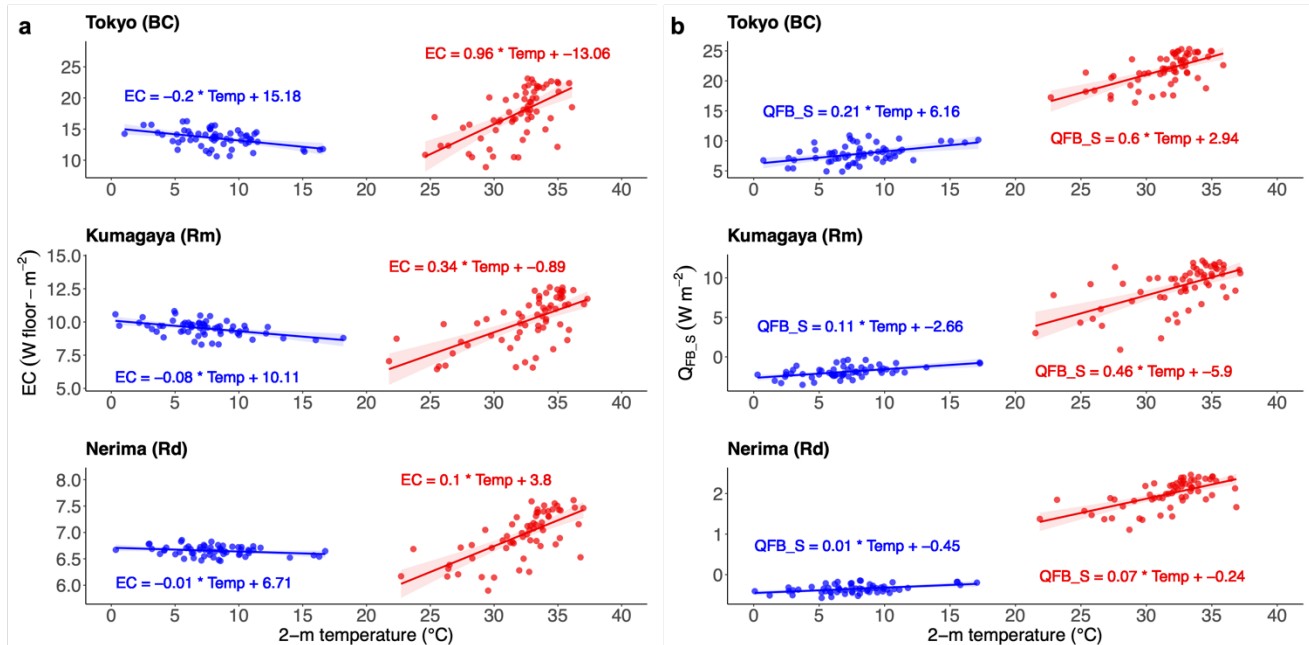

**Figure 10: Scatterplots of 2-m temperature and (a) electricity consumption (*EC*), and (b) anthropogenic sensible heat from buildings (*$Q_{FB\_S}$*) in Tokyo (BC), Kumagaya (Rm), and Nerima (Rd) at 14:00 LT in summer and winter simulated by SLUCM+BEM. Each plot shows daily results. Lines with error bars are single regression lines. Plots with temperatures > 20°C represent calculation results for summer; those with temperatures < 20°C represent calculation results for winter.**

**Table 3: The SLUCM+BEM- and CM-BEM-simulated *EC* temperature sensitivities ($\Delta EC/\Delta T$) and the observations at 14:00 LT during each season for all urban categories.**

|        |               | SLUCM+BEM | CM-BEM[1] | Observation[2] |
|--------|---------------|-----------|-----------|----------------|
| Summer | Tokyo (BC)    | 0.96      | 0.73      | 0.64           |
|        | Kumagaya (Rm) | 0.34      | -         | 0.25           |

| | | | | |
|---|---|---|---|---|
| | Nerima (Rd) | 0.1 | 0.48 | 0.29 |
| Winter | Tokyo (BC) | −0.20 | −0.01 | −0.41 |
| | Kumagaya (Rm) | −0.08 | – | −0.14 |
| | Nerima (Rd) | −0.01 | −0.13 | −0.17 |

[1] Nakajima et al. (2023), [2] Nakajima et al. (2022).

Like $EC$, $Q_{FB\_S}$ can be calculated in a temperature-dependent manner (Fig. 10b). As also noted for $EC$, $Q_{FB\_S}$ and temperature are positively correlated in summer. In this case, winter also shows a positive correlation due to the use of air-source air conditioning is used, leading to heat absorption (i.e., negative heat is emitted) from the outdoor air during heating. This heat absorption is more significant at lower outdoor temperatures.

Notably, in the original SLUCM, $EC$ is always zero, as it is not a target for calculation. The value of $Q_{FB\_S}$ does not respond
to air temperature (see Fig. 10). By contrast, in SLUCM+BEM, both $EC$ and $Q_{FB\_S}$ can be calculated to respond to air temperature. It is a significant achievement that these two variables can now be calculated dynamically after addressing the shortcomings of SLUCM.

## 4 Discussion

### 4.1 Importance of considering partial HAC

SLUCM+BEM includes features in the modelling of $EC$ and $Q_{FB}$ that are not considered in the BEP+BEM or officially included in the WRF, as follows.

· Consideration of partial HAC: BEP+BEM assumes that HAC is always in use on all floors and locations in the building, which is an unrealistic situation, and thus overestimates actual $EC$ and consequently $Q_{FB}$ emissions (Takane et al., 2017; Xu et al., 2018). To avoid this overestimation, this study introduced the concept of partial HAC (Section 2.1) as described previously (Takane et al., 2017).

· Consideration of changes in COP: In BEP+BEM, COP has a fixed input value. In practice, COP generally varies with ambient temperature. The consideration of changes in COP allows more realistic dynamic calculation of $EC$ and $Q_{FB}$.

· Consideration of the cooling tower: In BEP+BEM, all $Q_{FB}$ is emitted as sensible heat, irrespective of building use. However, cooling towers exist in offices, and some $Q_{FB}$ is discharged as latent heat during the cooling season, as

demonstrated by the detailed cooling tower model in BEP+BEM (e.g., Yu et al., 2019) and in our separately developed CM-BEM. Therefore, in SLUCM+BEM, simplicity is emphasised, and fractions are introduced in Eqs. (7) and (8) to reproduce a simple cooling tower.

This section discusses how each of these features affects the $Q_{FB\_S}$ output. The results for the control case, which considers all three of these items, are shown in Fig. 11a. $Q_{FB\_S}$ is more significant in central Tokyo and more minor in the suburbs. The temporal variations at three representative locations for each building use indicate that in Tokyo, $Q_{FB\_S}$ values increase after 06:00 and reaches 30 W m$^{-2}$ at around 11:00, peak at around 18:00, and then decrease. By contrast, in Kumagaya and Nerima, $Q_{FB\_S}$ values increase after 18:00, as more people are present in their houses at night than during the day. Thus, residential

areas use more AC at night than during the day (Table 2, AC_FLOOR_RATIO). Although the value of $Q_{FB\_S}$ is impossible to directly verify while considering all three of these factors, the calculation is regarded as realistic because it reproduced $EC$ well.

Figure 11b shows the difference when cooling towers were and were not (No cooling tower - CTRL) considered. As only offices feature cooling towers, the results for residential areas are similar to those obtained previously. When focusing only on

offices, the values for central Tokyo were more significant than those shown in Figure 11a. In terms of temporal variation in Tokyo, the $Q_{FB\_S}$ curve was the same as that described in the previous case, but the peak day value was over 40 W m$^{-2}$, higher than the peak of about 35 W m$^{-2}$ for the control scenario (Fig. 11a). Thus, cooling towers afforded an average day difference of approximately 15 W m$^{-2}$.

Next, we considered the effect of COP changes. Figure 11c shows the difference between a scenario that does not consider

COP changes (thus where COP is fixed ["No COP change"]) and a scenario with no cooling tower ("No COP change–No cooling tower). The effects of COP changes were less than those illustrated in Figure 11b. Figure 11c reveals almost no change in the $Q_{FB\_S}$ and that the temporal changes were near-identical at the three representative points. However, $Q_{FB\_S}$ changes should probably be considered when dealing with heat waves and as the urban climate becomes increasingly affected by global warming. The temperatures would then be significantly higher than those of the present study, lowering the COP and increasing

the $EC$ and $Q_{FB\_S}$ (Takane et al., 2019; 2020).

Finally, we considered the impact of partial HAC. We changed the settings of Figure 11c to incorporate a whole-of-house HAC (similar to BEP+BEM). We did not consider partial HAC use. Compared to the previous case, the $Q_{FB\_S}$ for the entire metropolitan area increased in the whole-of-house HAC scenario (Fig. 11d). The temporal changes at the three representative locations were also clearly affected. For example, in Tokyo, the nighttime $Q_{FB\_S}$ was greater for the whole-of-house HAC than

the partial HAC scenario, and the difference between the daytime and nighttime values smaller. $Q_{FB\_S}$ was approximately 90 W m$^{-2}$ regardless of the time of day. Kumagaya exhibited no significant variation in the diurnal pattern, but the absolute values were consistently above 40 W m$^{-2}$. In Nerima, the pattern shifted to a diurnal peak. Thus, consideration of partial HAC status

critically impacted our results. When including partial HAC in a model, new parameters such as those listed in Table 1 are needed to reflect accurately the effects of human activity. These (slightly) complicate the analysis. However, the difference
between the No partial HAC and No COP change scenarios (Fig. 11d) illustrates the need to consider partial HAC whenever possible; this strongly impacts the results. Social big data on the population, and electricity and HAC use, will be valuable. Such data were used by Takane et al. (2022) to establish the parameters described above.

Overall, these results suggest that all three of the features included in SLUCM+BEM, but not in BEP+BEM or WRF, for the modelling of $EC$ and $Q_{FB}$ should be considered. At a minimum, partial AC should be considered.

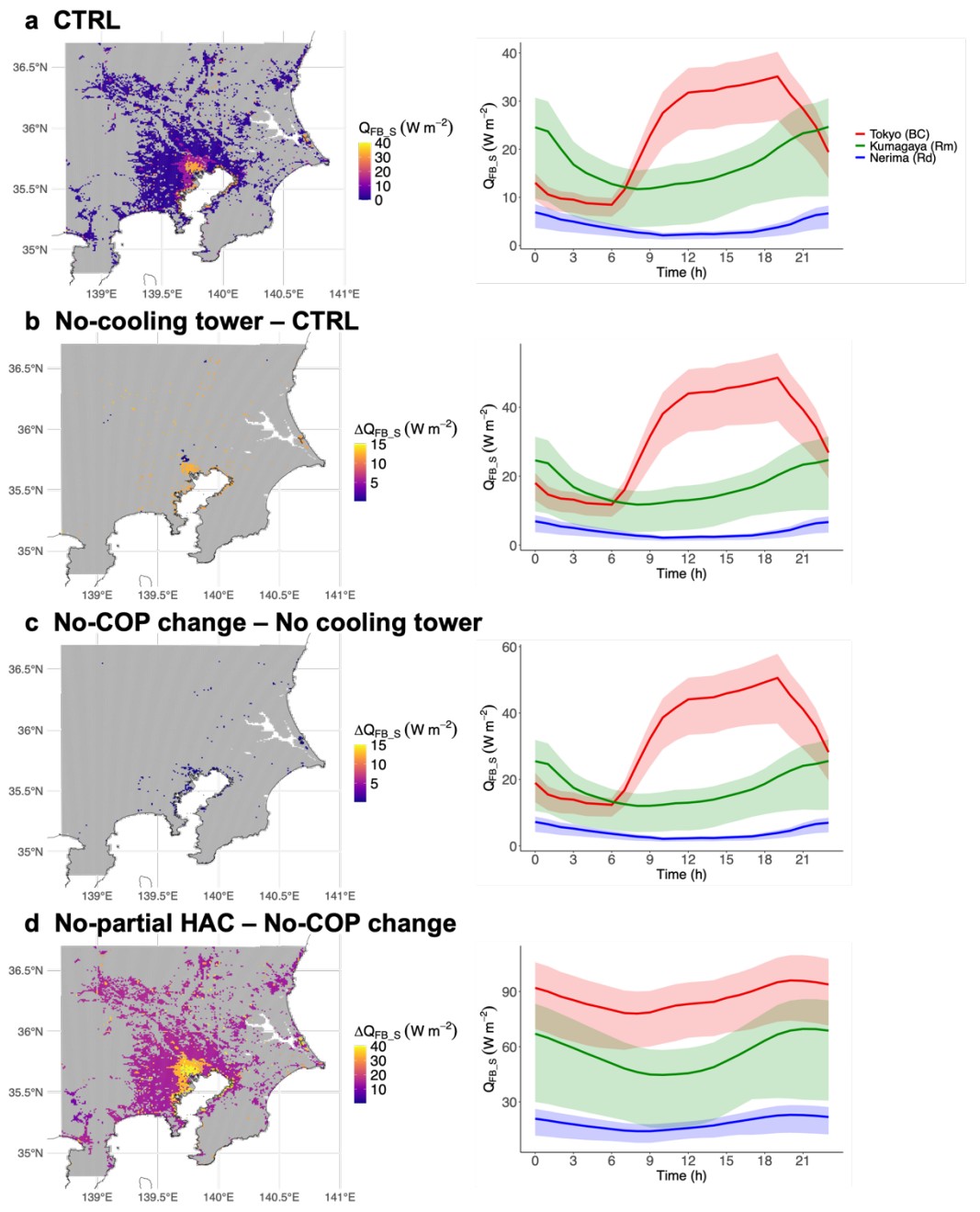

**Figure 11: The average, SLUCM+BEM-simulated $Q_{FB\_S}$ in distributions over the Tokyo Metropolitan Area averaged forat 14:00 LT in summer obtained from SLUCM+BEM (left). Diurnal changes in the $Q_{FB\_S}$ invalues for Tokyo (BC), Kumagaya (Rm), and Nerima (Rd) (right). Lines and error bars are the simulated average values and the 5th–95th percentiles, respectively. Simulation results areThe simulations were run for cases including (a) control (CTRL), (b) no cooling towerstower, (c) no coefficient of performance (COP) change, and (d) no partial HAC scenarios.**

**4.2 Guidance for model selection**

This section offers recommendations for model selection and the appropriate use of three urban models, SLUCM, SLUCM+BEM, and CM-BEM, each of which has different characteristics. An overview of the model selection process is provided in Fig. 12.

The most important difference affecting model selection is whether the user requires dynamic calculation of $Q_F$ and $EC$. If this calculation is not required, the original SLUCM is suitable for use. Notably, the two approaches to improving this model differ depending on whether BOUND* is set to 1 or 2 (see Sections 1 and 2.1). It is essential that $Q_F$ (AH, AHDIUPRF in URBPRAM.TBL) is entered as realistically as possible. If it is possible to enter realistic values for $Q_F$ obtained from energy consumption statistics compiled by the city or country of interest or from existing global databases (e.g., Varquez et al., 2021), then it is possible to reasonably simulate urban temperatures averaged over the simulation period (see Sections 1 and 2.1). For example, when BOUND* = 1 (zero-flux), the building is assumed to be perfectly insulated, whereas if $Q_F$ is entered separately and includes realistic values for heat removal from the building ($Q_{FB}$), then the calculation can be considered to reproduce realistic conditions. Similarly, when BOUND* = 2 (constant), the building acts as a heat sink or source at each time step, but if the energy lost or gained in this manner is added to $Q_F$ in advance, this calculation can also be considered to provide a realistic representation. In the case of constant, we recommend that the boundary conditions TRLEND and TBLEND are not set as the room temperature, but as the average outdoor temperature of the location during the calculation period. The reason for this setting is that entering the average outdoor temperature causes the calculation to assume that the energy balance between outdoors and indoors is approximately balanced, at least when averaged over the calculation period. This concept is similar to weather and climate simulations that use a bottom boundary condition of land-surface models.

Users who have difficulty in setting realistic values for $Q_F$ as described above, want to calculate $Q_F$ and $EC$ dynamically, or want to simulate a period with high temperature variations among days and time points are advised to use CM-BEM (or BEP+BEM as a model of the same type) and SLUCM+BEM. However, these two models also have different uses. Specifically, if $Q_F$ and $EC$ are required to be calculated in detail, such as considering a building in multiple vertical layers and calculating the heat load of the building including windows and ventilation, for realistic calculation of both $EC$ and gas consumption, or if rich input data related to these settings are available, then CM-BEM is an option.

If a single layer is sufficient instead of multi-layer analysis, if few input data are available, or if there are concerns about the $Q_F$ settings for SLUCM as described above, then the SLUCM+BEM proposed in this paper is the optimal choice. Notably, SLUCM+BEM is a parameterisation that assumes BOUND* =2 (i.e., constant) and the boundary conditions TRLEND and TBLEND assume the temperature setting of the HAC (room temperature), in contrast to the SLUCM constant setting. In our simulation environment (HPE Apollo 2000 [scalar computer], 3,072 GFlops, 192 GiB memory, Intel Xeon Gold 6148, 40-

core parallel computing, Intel compiler), the computation times for the entire SLUCM+BEM and SLUCM simulations were very similar.

As described above, SLUCM+BEM is a parameterisation that eliminates as many of the shortcomings of both SLUCM and
CM-BEM as possible, while incorporating as many of their benefits as possible. According to Chen et al. (2021), inadequate representation of building energy is included in many single-layer UCMs, including the surface urban energy and water balance scheme (SUEWS) (Järvi et al., 2011; 2014; Ward et al., 2016; Sun et al., 2024) and the Arizona State University single-layer urban canopy model (ASLUM) (Wang et al., 2013; Wang et al., 2021). The SLUCM improvement that we achieved via implementation of a simple BEM could be extended to other single-layer UCMs

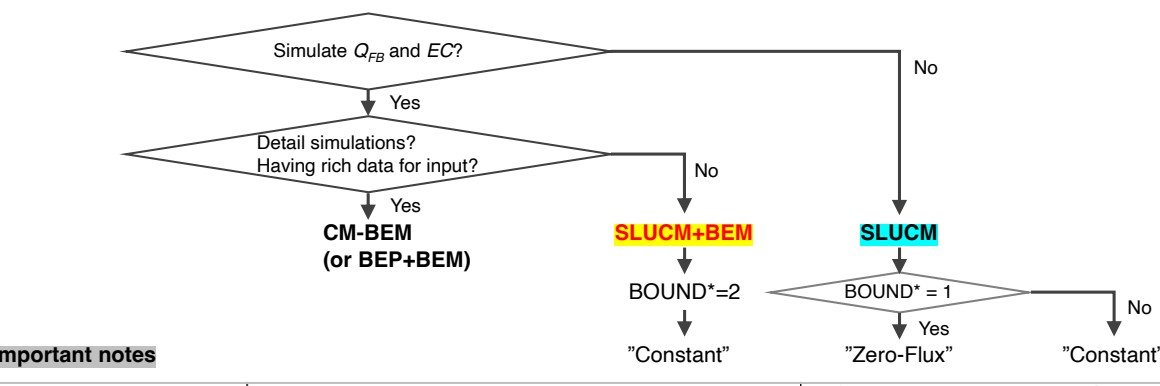

| Important notes | | | | | |
|---|---|---|---|---|---|
| **Assumed building conditions** | | **Real condition that conserve energy balance within the urban canopy in each time step** | | **Perfect insulation** | **Heat sink/source in each time step** |
| $Q_F$ | $Q_F$ from buildings ($Q_{FB}$) | Simulated | Simulated (AHOPTION = 2) | Input values as realistic as possible (AHOPTION = 1) | |
| | $Q_F$ from traffic | Input values as realistic as possible * | Input values as realistic as possible | | |
| $EC$ | $EC$ by HAC use | Simulated | Simulated | - | |
| | $EC$ by equipment | Input values as realistic as possible | Input values as realistic as possible | | |
| **Building related parameters** | Morphology | - Mean building width | - Normalised roof width | | |
| | | - Mean road width | - Normalised road width | | |
| | | - Distribution of building height | - Mean building height | | |
| | | - Window area | | | |
| | Heat insulating properties | - Building material in roof & walls | - Building material in roof & walls | | |
| | | - Heat insulating material in roof & walls | | | |
| | | - Window in walls | | | |
| **HAC related parameters** | Electricity | - Efficiency of HAC | - Efficiency of HAC | - | |
| | | - HAC usage fraction including their schedule | - HAC usage fraction including their schedule | | |
| | Gas | - Efficiency of HAC * | - | | |
| | | - HAC usage fraction including their schedule * | | | |
| **Boundary conditions** | TRLEND, TBLEND | - | Regards setting "indoor (room)" temperature by HAC | Default | Set "outdoor" temperature averaged by simulation period for conserving energy balance within the urban canopy during the period |

**Figure 12: Flowchart of model selection process, highlighting important features and conditions of each model.**

## 4.3 Limitations and future works

The factors that SLUCM+BEM ignores compared to the more detailed models BEP+BEM and CM-BEM are mainly windows and ventilation (Table 1). As no database of these factors exists at present, inaccurate window parameter inputs can lead to inaccurate calculation of indoor heat load, $EC$, and $Q_{FB}$. Therefore, we ignored these factors, because their inclusion deviates from the development policy of SLUCM+BEM, which was to develop the simplest model possible; we also ignored ventilation for the sake of simplicity. We show here how ignoring these processes affects the total indoor heat load $H_{in}$. We use the results of the CM-BEM model that takes such processes into account. Table 4 shows the contributions of windows (specifically, insulation of solar radiation [SR] through windows) and ventilation (sensible heat exchange [VENT]) to $H_{in}$. During a summer day, SR and VENT attain +15.3 W floor-m$^{-2}$ and –7.6 W floor-m$^{-2}$ respectively, resulting in a net sensible heat gain of +7.7 W floor-m$^{-2}$. SLUCM+BEM underestimates this +7.7 W floor-m$^{-2}$ (about 25% of $H_{in}$). However, CM-BEM tends to overestimate the daytime indoor temperature compared to the observations, suggesting that CM-BEM may also overestimate $H_{in}$. This suggestion is supported by the $EC_{HAC}$ overestimations at the BC grids of Figure 7. Such overestimations are in part explained by the fact that CM-BEM does not consider blinds, which are of course common in offices and residential buildings. Thus, the figure of +7.7 W floor-m$^{-2}$ may be an overestimate. At night, the SR and VENT are +0.5 W floor-m$^{-2}$ and –6.4 W floor-m$^{-2}$ respectively, resulting in a net sensible heat gain of –6.0 W floor-m$^{-2}$. Thus, the SLUCM+BEM overestimate is about 6.0 W floor-m$^{-2}$. During a winter day, SR and VENT attain +17.3 W floor-m$^{-2}$ and –15.0 W floor-m$^{-2}$ respectively, resulting in a net sensible heat gain of +2.3 W floor-m$^{-2}$, thus lower than in summer. At night, SR and VENT are 0.0 W floor-m$^{-2}$ and –16.0 W floor-m$^{-2}$ respectively; the net sensible heat gain is –16.0 W floor-m$^{-2}$. Therefore, SLUCM+BEM may overestimate $H_{in}$. In addition, SLUCM+BEM does not consider dehumidification, which contributes to $H_{in}$. Simple inclusion of such processes is desirable in future research when a good global dataset related these are available.

Table 4: The contributions of processes that SLUCM+BEM ignores: The effects of SR and VENT on $H_{in}$ simulated by CM-BEM during the days and nights of each season.

| | | $H_{in}$ [W floor-m$^{-2}$] | SR [W floor-m$^{-2}$] | VENT [W floor-m$^{-2}$] | SR–VENT (net sensible heat gain) [W floor-m$^{-2}$] |
|---|---|---|---|---|---|
| Summer | Daytime | +31.5 | +15.3 | –7.6 | +7.7 |
| | Nighttime | -10.1 | +0.5 | –6.5 | –6.0 |
| Winter | Daytime | +5.9 | +17.3 | –15.0 | +2.3 |
| | Nighttime | –48.3 | 0.0 | –16.0 | –16.0 |

$H_{in}$, indoor sensible heat load; SR, solar radiation insolation through windows; VENT, sensible heat exchange afforded by ventilation.

In addition, SLUCM+BEM considers only sensible heat. The balance of latent heat within and outside the building and the latent heat content of $Q_{FB}$ are not calculated dynamically, in contrast to BEP+BEM and CM-BEM.

Another limitation of SLUCM+BEM is that the model considers that the boundary wall and roof temperatures (TBLEND and
535 TRLEND) set the room temperature for the HAC system. This aids simplification, but may cause $EC_{HAC}$ to be overestimated (Oleson & Feddema, 2020). In detail, TBLEND and TRLEND are usually higher/lower than the room temperature in summer/winter. Therefore, the use of TBLEND and TRLEND to set the room temperature requires more energy (Oleson & Feddema, 2020); $EC_{HAC}$ is potentially overestimated. We tried to avoid this by setting the temperatures slightly higher/lower for the summer/winter simulations (Table 2). However, it is important, in future, to model the room temperature with
540 consideration of convective and radiative heat exchange between the interior wall and roof, and indoor air, as in previous works (Kikegawa et al., 2003; Oleson & Feddema, 2020).

Furthermore, like BEP+BEM, SLUCM+BEM assumes weekday patterns for all calculations and does not consider weekends, whereas CM-BEM does differentiate weekends (Table 1). This change can lead to temperature differences of approximately 0.1–0.6°C in urban centres, particularly on holidays (Fujibe, 1987; 2010; Bäumer & Vogel, 2007; Ohashi et al., 2016; Earl et
al., 2016). This limitation may have led to an overestimation of $EC_{HAC}$ in BC, as described in Section 3.2.2. Nevertheless, the number of holidays is limited compared to weekdays, and in this study, avoiding complexity was prioritised over this effect.

The most challenging point in parameterising $Q_{FB}$ and $EC$ is the treatment of heating. In Japan, air-source heat pump AC units are also used for heating, but heating represents a smaller percentage of their use than cooling (Takane et al., 2017; 2023). No accurate data on the actual percentage of their service is available. Despite a trend toward using heat pump AC units for heating
in other countries, particularly in the EU, this practice is not yet common. Therefore, winter calculations should be conducted with more caution than summer calculations. We must emphasise that the same limitation and caution must be applied for existing models such as BEP+BEM. In addition, parameterisation based on the air-source heat pump AC will become increasingly important in future scenarios. Heat pumps aid decarbonisation and, thus, are attracting increasing attention. Such pumps will become widely used to ensure energy security. By contrast, CM-BEM considers heating types other than air-source
heat pump AC (e.g., Kikegawa et al., 2003). Nonetheless, this CM-BEM setting is too complex for meteorologists and climatologists, who are the main users of WRF, and the data on which this setting is based are not standard. SLUCM+BEM avoids this complexity.

The SLUCM+BEM did not focus on urban hydrological processes such as biophysical and echophysiological characteristics of roof and ground vegetation and urban trees. However, these processes play an important role in the energy balance of the
560 urban canopy (e.g. Lemonsu et al., 2012; Krayenhoff et al., 2020; Meili et al., 2020). Implementation and evaluation of these processes is another future work.

The BEM developed in this study shares certain challenges with other BEMs. Although the BEM can accurately calculate the temporal variation and spatial distribution of anthropogenic heat emissions, it may not correctly calculate their long-term average values and spatial averages. This issue is reminiscent of the shortcomings of the bottom-up approach used to create anthropogenic heat emission databases from statistical data for energy consumption amounts. When creating anthropogenic heat emission databases, this problem could be addressed by concurrently employing a top-down approach, in which anthropogenic heat emission data are calculated based on a statistical energy consumption database. Users of the BEM may address this issue by skillfully adjusting parameters while verifying the estimated anthropogenic heat against statistical data.

In general, if the information input to the model (optimal input data, parameter settings) is insufficient, a more sophisticated model will have worse accuracy. In other words, there is an inextricable link between the information input to the model and the accuracy of the simulation results (e.g., Takane et al., 2023b). Therefore, users should carefully consider the information available for their target city and select a model that is appropriate for that information. In addition, the most important method for improving the accuracy of the model may be the development of urban information, including morphological parameters (e.g., Khanh et al., 2023) and social big data such as real-time population and energy consumption data (e.g., Takane et al., 2023b), which can effectively exploit the potential of a sophisticated model such as BEM.

Future studies will include the projection of $Q_{FB}$ emissions, $EC$, and urban climates under future climate conditions, direct comparison with BEP+BEM, addressing the local climate zone (Demuzere et al., 2022), and application to cities other than Tokyo.

**5 Summary**

The SLUCM, which has many users worldwide, has limitations including constant anthropogenic heat ($Q_F$) and fully adiabatic conditions or energy imbalance within the urban canopy layer in each time step. The present study addressed these limitations through developing a new dynamic parameterisation: SLUCM+BEM. The development philosophy underlying this parameterisation and its usage is summarised as follows.

To maintain the simplicity that is the major advantage of SLUCM, we addressed its limitations as simply as possible and proposed a dynamic parameterisation of electricity consumption ($EC$) and $Q_F$ from buildings ($Q_{FB}$), designated SLUCM+BEM. To address the limitations of SLUCM, the most critical process was calculating conductive heat transfer, from which $EC$ and $Q_{FB}$ are calculated. In doing so, windows and ventilation are not considered for the sake of simplicity.

The input parameters for BEP+BEM (HSEQUIP_SCALE_FACTOR and HSEQUIP) are re-used for the calculations outlined above, and five new parameters are incorporated into URBPRAM.TBL. The implementation of SLUCM+BEM is simple. Specifically, realistic values are set for the new parameters, and AHOPTION is set to 2 in URBPRAM.TBL.

Using the proposed settings, SLUCM+BEM reproduced the radiation balance and surface heat budget within the urban canopy layer at Tokyo (Yoyogi) in summer (cooling season) and winter (heating season) as well as SLUCM. SLUCM+BEM reproduced the temporal variation and spatial distribution of air temperature in summer (cooling season) and winter (heating season) as well as SLUCM.

The development of SLUCM+BEM enables the dynamic calculation of $EC$ and $Q_{FB}$. SLUCM+BEM provided good representation of the temporal variation and spatial distribution of $EC_{HAC}$ in summer (cooling season) and winter (heating season). Compared to the more sophisticated model CM-BEM, SLUCM+BEM less accurately reproduced the fine spatial distribution in urban areas, particularly in BC grids. However, SLUCM+BEM showed similar accuracy to CM-BEM in reproducing spatially averaged values, particularly in summer. The reproducibility of $EC$ suggests that $Q_{FB}$ calculated from
$EC$ is also fairly realistic.

SLUCM+BEM introduces several processes (i.e., partial HAC, COP changes, and cooling towers) that are not considered in the official BEP+BEM. Of these processes, the consideration of partial HAC is most critical, as it significantly affects the value of $Q_{FB}$. Therefore, it is essential to introduce the five new parameters as accurately as possible.

The computation times for the entire SLUCM+BEM and SLUCM simulations were very similar.

The source code for SLUCM+BEM has been made openly available (Takane et al., 2024b); thus, it may be freely accessed by WRF and SLUCM users.

**Code and data availability**

All datasets analysed in this work are publicly available. The WRF model may be downloaded at https://github.com/wrf-model (last accessed: 11/09/2023). The input data and source code for WRF-SLUCM+BEM used in this study have been archived
on Zenodo at https://doi.org/10.5281/zenodo.10685693 (Takane et al., 2024a) and https://doi.org/10.5281/zenodo.10686465 (Takane et al., 2024b), respectively.

**Author contribution**

YT and YK designed the study and YT led the development of WRF–SLUCM+BEM with significant contributions from YK and HK. YT and KN performed the evaluation. YT, YK and HK drafted the manuscript, and all authors reviewed and edited
the manuscript.

**Competing interests**

The authors declare that they have no conflict of interest.

**Acknowledgements**

We thank two anonymous reviewers for their valuable comments, which helped us to improve the manuscript. We also thank
Dr Hirofumi Sugawara for providing the radiation and heat flux data observed at the Yoyogi site in Tokyo. This study was supported by the Environmental Research and Technology Development Fund (grant no. JPMEERF20231007) of the Environmental Restoration and Conservation Agency of Japan. We were also supported by  Japan Society for the Promotion of Science (JSPS) KAKENHI grant (no. JP23H01544). The calculations were performed using the supercomputer system (NEC SX-Aurora TSUBASA) of the National Institute for Environmental Studies. We thank Dr. Masayuki Hara of the Japan
Meteorological Agency for his technical support using the LULC datasets of the Geospatial Information Authority of Japan for WRF simulations.

**Financial support**

This study was supported by the Environmental Research and Technology Development Fund (grant no. JPMEERF20231007) of the Environmental Restoration and Conservation Agency of Japan and Japan Society for the Promotion of Science (JSPS)
KAKENHI grant (no. JP23H01544).

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
