# Peer review of "SLUCM+BEM (v1.0): A simple parameterisation for dynamic anthropogenic heat and electricity consumption in WRF-Urban (v4.3.2)"

_EGUsphere, 2024_

## Author Comment (AC2)

We thank the reviewers and the editor for the comments on our manuscript.
In the following, the format is:

      *Reviewer's comment*

      Our reply (revised text and place)

**Reviewer #1**

*General comments*

*This paper by Takane et al. describes the model development efforts of adding a building energy model (BEM) into the single-layer urban canopy model (SLUCM) coupled with the Weather Research and Forecasting (WRF) model. I think it represents a significant advancement in WRF-SLUCM model development that will help expand the usage of WRF-SLUCM and make building energy-urban climate interaction studies more accessible for those with limited computational resources. However, I believe the following concerns need to be addressed before this paper can be published.*

Thank you very much for your review and the positive comments. We have made the suggested revisions.

*Specific comments*

*My major concern involves the apparent overestimation of HAC energy use as shown in Fig. 6 and Fig. 8, which is not clearly acknowledged, and the reason for this overestimation is insufficiently addressed. There are some key assumptions made in developing the simple BEM that may have affected the simulated HAC energy use, but their impacts are not discussed.*

*First, the impact of the neglected parameters (solar heat gain through windows, sensible heat gain through ventilation, and the latent heat load from dehumidification in summer) on the simulated HAC energy use was not addressed. While I agree with the author's choice of not parameterizing these heat transfer processes based on the principle of keeping the model as simple as possible and to avoid the uncertainty introduced by not having good-quality data necessary to accurately parameterize these processes, I do think the effect of not including these factors should be acknowledged and analyzed.*

*Specifically, ignoring direct solar heat gain through windows should mean that the space heating in winter is likely overestimated and space cooling in summer underestimated (Sailor, 2011). Not considering ventilation should lead to an underestimation in both heat and cooling, whereas not considering dehumidification would also underestimate cooling energy consumption, especially for a city like Tokyo which has humid summers. The effects of these factors, when combined, compensate each other for heating energy use (i.e., the simulated heating energy use could be underestimated or overestimated as a result of ignoring these*

*processes), but should lead to an underestimated cooling energy use. However, from Fig. 6, we*
*see the SLUCM-BEM simulated EC_HAC in summer is almost universally higher than*
*observations, i.e., SLUCM-BEM overestimates EC_HAC despite all the neglected factors which*
*should actually lead to an underestimated EC_HAC. This suggests that the newly developed*
*BEM likely has compensating factors that significantly overestimates summer EC_HAC, which*
*leads to my second point of concern, as the reason for this overestimation is not sufficiently*
*addressed in the discussion.*

*The authors discussed one possible cause of this overestimation which is that SLUCM-BEM does*
*not consider weekday-weekend differences (L680 - 681). While this may explain the*
*overestimation in the BC grids, it does not explain the overestimation in the residential districts.*
*I would be interested to see the SLUCM-BEM results validated against the observation data for*
*weekday only. If the overestimation is still there, it is an indication that other factors are at play.*

Thank you. In response, we have made the three revisions described below; we recalculated the
results; and we replaced all results of the original manuscript.

Revision points:
1. When we checked Table 2 (urban parameter settings), we found that the internal heat gain
   value ("HSEQUIP_SCALE_FACTOR") of SLUCM+BEM was much larger than that of
   CM-BEM. We therefore adjusted this value to match that of CM-BEM; this allows
   comparison. The other parameters have been slightly modified.
2. We found an error in the source code. Specifically, we used $H_{in}$ instead of $H_{out}$ when
   originally calculating $EC_{HAC}$. In other words, the numerator on the right of equation (4) was
   $H_{in}$. Note that this error did not affect other variables, although it was critical for $EC_{HAC}$
   simulated in the original manuscript. This has been corrected ("bug fix").
3. We revised the set temperature by HAC (TBLEND and TRLEND) in line with the comment
   of reviewer #1: "*More energy is required to keep indoor surfaces at the setpoints than to*
   *keep air temperature at the setpoints.*". This means that "the set temperatures in this paper
   (the interior wall surface temperature [TBLEND] and the roof surface temperature
   [TRLEND]) and the actual room temperature (Tin) differ in reality (summer: TBLEND
   (TRLEND) > Tin, and winter: TBLEND (TRLEND) < Tin)". Thus, the set temperatures in
   the simulation have been revised.
      When revising the set temperatures (the difference between TBLEND [TRLEND] and
   Tin), it is undeniable that some "tuning" is in play. This must be addressed in future, as must
   the modelling of room temperature, which we discuss below.

The recalculations after making the above corrections are shown in Figure R1. The corrections
significantly improved the results.

In addition, an error analysis was performed for weekdays only. We explored the effects of
holidays (as mentioned in the original manuscript). When only weekdays were compared, the
overestimation was further (slightly) improved (revised [all days] vs. revised [weekdays]) (Fig.
R1).

| | |
|---|---|
| Original (all days) |
[Figure]
 |
| Revised (all days) | |
| Revised (weekdays) | |

**Figure R1: Diurnal changes in (a) the MBE and (b) the MAE of $EC_{HAC}$ for each type of urban building, thus Rm, Rd, and BC; and the summer averages for all SLUCM+BEM grids given in the original manuscript (all days) (upper panels); in the revised manuscript (all days) (middle panels); and in the revised manuscript only for weekdays (lower panels).**

In contrast, for the CM-BEM, which originally considered the difference between weekdays and holidays, the weekday error did not improve more than did the error for all days (no Figure shown). The original manuscript showed the error for all days, including holidays, but the revised manuscript shows the error for weekdays only.

The revised text reads:

"We focused on validation of $EC_{HAC}$; this is the variable simulated by the models. The observed $EC_{HAC}$ was that estimated by Nakajima et al. (2022). It is better to validate $EC_{HAC}$ rather than EC because $EC_{HAC}$ is the actual simulated variable; $EC$ includes input baseload parameters (HSEQUIP_SCALE_FACTOR and HSEQUIP). Thus, the $EC$ validation contains errors in both the simulated $EC_{HAC}$ and the input parameters. Nakajima et al. (2022) showed that the baseload tended to vary even among central Tokyo BC grids of the same category. CM-BEM considers baseload variability because CM-BEM inputs different baseload values into each model grid, whereas SLUCM+BEM employs only one baseload for each urban category (the input is thus uniform across all BC grids; Table 2). Therefore, we focused only on $EC_{HAC}$ when comparing the simulated variables of SLUCM+BEM and CM-BEM. The verification focused only on the weekdays of the simulated period; the SLUCM+BEM considers only weekday conditions, as does BEP+BEM.

Figure 6a is a detailed map of the Tokyo metropolitan $EC_{HAC}$ in summer (July–August 2018 weekday average) as presented by Nakajima et al. (2023) and Takane et al. (2023). Figure 6b is focused on central Tokyo. $EC_{HAC}$ is higher in the city centre and decreases toward the suburbs; SLUCM+BEM generally captured this (city centre > suburbs) (Fig. 6c, d vs. a, b). The $EC_{HAC}$ errors by the building type, and time, within the areas of Figure 6b and d are shown in Figure 7 (upper panel). In Rm residential grids, the daily mean bias error (MBE) was 0.8 W floor-m$^{-2}$ and the MAE 1.5 W floor-m$^{-2}$. The Rd residential grids exhibited slightly better results, with a daily MBE of –0.8 W floor-m$^{-2}$ and an MAE of 1.3 W floor-m$^{-2}$. In contrast, BC grids yielded a daily MBE of 2.8 W floor-m$^{-2}$ and an MAE of 3.5 W floor-m$^{-2}$; the errors were greater than those of the residential grids. $EC_{HAC}$ tended to be high after 11:00 LT. Despite overestimation of the BC grids, the total, daily average errors for the areas shown in Figure 6b and d were MBE = –0.1 W floor-m$^{-2}$ and MAE = 1.5 W floor-m$^{-2}$, because the BC grid area was smaller than those of the Rm and Rd grids (Fig. 2).

The results obtained using a more detailed model, thus CM-BEM (Kikegawa et al., 2003; 2014, 2022; Takane et al., 2022; Nakajima et al., 2023) are compared with the SLUCM+BEM data in Figure 6e and f. The CM-BEM results cover a limited area; the computational coverage is low compared to that of SLUCM+BEM. Although the areas for which $EC_{HAC}$ were calculated differ, the model resolutions (1 km) and physical parameterisations are identical, except for those of the urban canopy and building energy models. Comparisons are possible. The CM-BEM results (Fig. 6f) well-reproduced the observations (Fig. 6b). In particular, SLUCM+BEM yielded a relatively uniform BC $EC_{HAC}$ for the city centre. In contrast, the CM-BEM values differed for each grid, in good agreement with the observations. The BC errors of CM-BEM and SLUCM+BEM were comparable; the daily MBE was 2.1 W floor-m$^{-2}$ and the MAE 2.5 W floor-m$^{-2}$. For the Rm residential grids, the daily mean errors were MBE = 0.8 W floor-m$^{-2}$ and MAE = 1.2 W floor-m$^{-2}$ (Fig. 7, bottom panel). As for the SLUCM+BEM data, the Rd residential results were slightly better than the Rm results, with daily mean errors of MBE = 0.4 W floor-m$^{-2}$ and MAE = 1.0 W floor-m$^{-2}$. As shown in Figure 6b and f, the daily average errors were MBE = 0.7 W floor-m$^{-2}$ and MAE = 1.2 W floor-m$^{-2}$, thus similar to those of SLUCM+BEM. Thus, although SLUCM+BEM is simpler than CM-BEM and can cover a larger area, it performed as well as did the detailed CM-BEM when validating $EC_{HAC}$ over the entire target area."

[Figure]

**Figure 7: Diurnal changes in (a) the MBE and (b) the MAE of $EC_{HAC}$ for each urban building type, Rm, Rd, and BC, and the average of all grids of the SLUCM+BEM (upper panels) and the CM-BEM (new model; lower panels) averaged over summer weekdays.**

In addition, we employed the CM-BEM to evaluate the contributions to the indoor heat load ($H_{in}$) that SLUCM+BEM ignores. CM-BEM considers all processes except the latent heat load attributable to dehumidification.

Lines 584–598 in the revised manuscript read:
"We show here how ignoring these processes affects the total indoor heat load $H_{in}$. We use the results of the CM-BEM model that takes such processes into account. Table 4 shows the contributions of windows (specifically, insulation of solar radiation [SR] through windows) and ventilation (sensible heat exchange [VENT]) to $H_{in}$. During a summer day, SR and VENT attain +15.3 W floor-m$^{-2}$ and –7.6 W floor-m$^{-2}$ respectively, resulting in a net sensible heat gain of +7.7 W floor-m$^{-2}$. SLUCM+BEM underestimates this +7.7 W floor-m$^{-2}$ (about 25% of $H_{in}$). However, CM-BEM tends to overestimate the daytime indoor temperature compared to the observations,

suggesting that CM-BEM may also overestimate $H_{in}$. This suggestion is supported by the $EC_{HAC}$ overestimations at the BC grids of Figure 7. Such overestimations are in part explained by the fact that CM-BEM does not consider blinds, which are of course common in offices and residential buildings. Thus, the figure of +7.7 W floor-m$^{-2}$ may be an overestimate. At night, the SR and VENT are +0.5 W floor-m$^{-2}$ and –6.4 W floor-m$^{-2}$ respectively, resulting in a net sensible heat gain of –6.0 W floor-m$^{-2}$. Thus, the SLUCM+BEM overestimate is about 6.0 W floor-m$^{-2}$. During a winter day, SR and VENT attain +17.3 W floor-m$^{-2}$ and –15.0 W floor-m$^{-2}$ respectively, resulting in a net sensible heat gain of +2.3 W floor-m$^{-2}$, thus lower than in summer. At night, SR and VENT are 0.0 W floor-m$^{-2}$ and –16.0 W floor-m$^{-2}$ respectively; the net sensible heat gain is –16.0 W floor-m$^{-2}$. Therefore, SLUCM+BEM may overestimate $H_{in}$. In addition, SLUCM+BEM does not consider dehumidification, which contributes to $H_{in}$. Simple inclusion of such processes is desirable in future research when a good global dataset related these are available."

**Table 4: The contributions of processes that SLUCM+BEM ignores: The effects of SR and VENT on $H_{in}$ simulated by CM-BEM during the days and nights of each season.**

|  |  | $H_{in}$ [W floor-m$^{-2}$] | SR [W floor-m$^{-2}$] | VENT [W floor-m$^{-2}$] | SR–VENT (net sensible heat gain) [W floor-m$^{-2}$] |
|---|---|---|---|---|---|
| Summer | Daytime | +31.5 | +15.3 | –7.6 | +7.7 |
|  | Nighttime | -10.1 | +0.5 | –6.5 | –6.0 |
| Winter | Daytime | +5.9 | +17.3 | –15.0 | +2.3 |
|  | Nighttime | –48.3 | 0.0 | –16.0 | –16.0 |

$H_{in}$, indoor sensible heat load; SR, solar radiation insolation through windows; VENT, sensible heat exchange afforded by ventilation.

*I suspect another possible (perhaps more important) cause of this overestimation may be due to another assumption that SLUCM+BEM made: when computing H_in, SLUCM+BEM assumes constant interior wall and roof surface temperatures (TBLEND and TRLEND), which essentially functions as "HAC setpoints", since a certain proportion of H_in will be removed/added during cooling/heating to maintain constant indoor surface temperatures. No indoor heat transfer processes (such as convection and radiation) are considered. This deviates from reality, as HAC setpoints dictate a constant indoor air temperature, rather than constant surface temperatures. More energy is required to keep indoor surfaces at the setpoints than to keep air temperature at the setpoints. This overestimation is also reflected in CLMU4 where a similar treatment of indoor temperature was employed (Oleson 2012; Oleson and Feddema, 2020). The authors should examine the implication of this assumption and, if at all possible, show the impact of this assumption on the simulated EC_HAC with simulations.*

*In summary, the key assumptions made in developing SLUCM+BEM model and their impacts should be thoroughly discussed so that future users are aware of these limitations when they interpret the results from the model.*

We agree that the temperatures of the inner walls and roof (TBLEND and TRLEND respectively) differ from the room temperature (Tin). Therefore, as mentioned above, we reviewed and recalculated the set temperature. The reviewer correctly emphasises the importance of modelling Tin as the heat exchange mediated via convection and radiation from the walls and roof. We attempted to model Tin using the CM-BEM method. Specifically, we employed a current offline model (standalone SLUCM+BEM) to calculate the room temperature variations over time by reference to heat conduction (radiation and convection) from the interior side walls and ceilings when an HAC system was used. The Figure below shows an example. Tin changes over time (Fig. R2, top left). It is now possible to qualitatively confirm that Tin changes as HAC usage varies.

[Figure]

**Figure R2: Diurnal changes in Tin_URB (indoor temperature), TH2_URB (outdoor temperature), QFB_URB (anthropogenic heat from a building), Hin_URB (the indoor sensible heat load), COP_URB (the coefficient of performance), and Hout_URB (the sensible heat load processed by HAC systems) derived via an improved SLUCM+BEM offline simulation of the Tokyo residential grid for July-August 2018.**

However, it is difficult to quickly (within the allowed revision period) implement this improvement in the online WRF model in a bug-free manner. We would like to describe the

improvement in a future paper. We now note that the current SLUCM+BEM model has the limitations described above.

Lines 606–613 of the revised manuscript:
"Another limitation of SLUCM+BEM is that the model considers that the boundary wall and roof temperatures (TBLEND and TRLEND) set the room temperature for the HAC system. This aids simplification, but may cause $EC_{HAC}$ to be overestimated (Oleson & Feddema, 2020). In detail, TBLEND and TRLEND are usually higher/lower than the room temperature in summer/winter. Therefore, the use of TBLEND and TRLEND to set the room temperature requires more energy (Oleson & Feddema, 2020); $EC_{HAC}$ is potentially overestimated. We tried to avoid this by setting the temperatures slightly higher/lower for the summer/winter simulations (Table 2). However, it is important, in future, to model the room temperature with consideration of convective and radiative heat exchange between the interior wall and roof, and indoor air, as in previous works (Kikegawa et al., 2003; Oleson & Feddema, 2020)."

Thank you for your understanding.

*Other comments/questions:*

- *The claim in the abstract, "Our results demonstrate that SLUCM-BEM can be applied to urban climates worldwide", seems inadequately supported. The authors only simulated and presented results for Tokyo, which seems insufficient to claim the applicability of this model worldwide. I understand the authors might be saying that, because this simpler model requires fewer input parameters, it has better potential to be applied to cities globally than other more complicated models. If that is the case, the author should make it clear, as the current sentence might give readers the impression that SLUCM-BEM is ready to be applied globally, whereas in reality it stills requires collecting local data on many parameters (e.g., HSEQUIP_SCALE_FACTOR, HSEQUIP, AB_BUILD_RATIO, AC_FLOOR_RATIO) to be able to accurately simulate HAC energy use.*

Thank you. We agree that many parameters would be required were the model to be applicable worldwide. This is a future work. We have deleted the text.

- *The definition of anthropogenic heat is a bit unclear to me. From Eqs. 6 and 7, this included H_out, which partially comes from the conductive heat transfer through walls and roofs. This part of the energy is already in the climate systems, which is not "anthropogenic" by nature. Anthropogenic heat fluxes (AHF) datasets are also derived based on non-renewable primary energy consumption (e.g., Flanner, 2009; Varquez et al. 2021). The authors should clarify if they are using a definition different from those used in the AHF datasets to avoid confusion.*

We define $Q_{FB}$ as anthropogenic heat from buildings. This includes the $H_{out}$ of equations (6) and (7). This definition differs from that of the AHF datasets. We have added text to the revised manuscript:
Lines 214–216:

"Note that the $Q_{FB}$ simulated by SLUCM+BEM is the anthropogenic heat from buildings. This includes the $H_{out}$ of equations (6) and (7). This definition differs from that of the anthropogenic heat flux (AHF) datasets that are focused on non-renewable, primary energy consumption (e.g. Flanner, 2009; Varquez et al., 2021)."

- *It is intriguing to me that SLUCM+BEM (also CM-BEM in winter) is unable to reproduce the EC diurnal profile. Any thoughts on possible causes?*

Reproduction of the daily $EC_{HAC}$ changes in summer and winter.
We could not reproduce $EC_{HAC}$ daily changes because, as mentioned above, we mistakenly used $H_{in}$ when calculating $EC_{HAC}$. This does not reflect the HAC usage schedule. $H_{out}$, (which we should have used), does. Also, the HAC use schedules varied. Although recent attempts have been made to determine the schedules using big social data (Takane et al. 2022; Nakajima et al. 2023), such data are not entirely adequate yet.

Winter.
Central heating is the most common form of heating in Europe and North America. In Japan, the means of heating are more diverse. Specifically, HAC may be locally combined with heaters or a kotatsu (a small table with an electric heater underneath, covered by a blanket), or HAC may not be used at all. Although we sought to capture different heating methods as simply as possible using certain parameters (AC_USAGE_RATIO_HT, AC_FLOOR_RATIO), we do not know how often air conditioning units are used for heating. This affects the winter $EC_{HAC}$ values of both SLUCM+BEM and CM-BEM. Please see Section 4.3.

- *In Fig. 10, the simulated EC-T sensitivities are presented. How do they compare with observations from Nakajima et al. 2022?*

We have added a new Table 3 and, in the text, we compare the data with observations (Nakajima et al. 2022) and the CM-BEM information (Nakajima et al. 2023).

Lines 457–467:
"The $EC_{HAC}$ calculation described above depends on the ambient temperature. The relationships between $EC$ and air temperature at representative locations in Tokyo (BC), Kumagaya (Rm), and Nerima (Rd) are shown in Figure 10a. In summer, the $EC$ and the temperature were positively correlated; the slope of the regression line indicates the temperature-sensitivity of $EC$ ($\Delta EC/\Delta T$). Conversely, the correlation is negative in winter, and the regression line slope shallower than in summer, in part because fewer buildings use air conditioning for heating in winter than for cooling in summer (e.g. Takane et al., 2017).
    The signs of the $\Delta EC/\Delta T$ values calculated by SLUCM+BEM were the same as those of the observations (positive in summer and negative in winter). The $\Delta EC/\Delta T$s simulated by SLUCM+BEM for summer are slightly overestimate in BC and Rm and underestimate in Rd, but these are reasonably good with observation (Table 3). In contract, the simulated values in winter tended to be smaller than the observations regardless urban category (Table 3). CM-BEM has the same feature as SLUCM+BEM; CM-BEM is reasonably good in summer but tended to

underestimate $\Delta EC/\Delta T$ in winter. It is important to improve the $\Delta EC/\Delta T$ by SLUCM+BEM and CM-BEM especially in winter. This is a future challenge."

**Table 3: The SLUCM+BEM- and CM-BEM-simulated *EC* temperature sensitivities ($\Delta EC/\Delta T$) and the observations at 14:00 LT during each season for all urban categories.**

|        |                 | SLUCM+BEM | CM-BEM[1] | Observation[2] |
|--------|-----------------|-----------|-----------|----------------|
| Summer | Tokyo (BC)      | 0.96      | 0.73      | 0.64           |
|        | Kumagaya (Rm)   | 0.34      | -         | 0.25           |
|        | Nerima (Rd)     | 0.1       | 0.48      | 0.29           |
| Winter | Tokyo (BC)      | –0.20     | –0.01     | –0.41          |
|        | Kumagaya (Rm)   | –0.08     | –         | –0.14          |
|        | Nerima (Rd)     | –0.01     | –0.13     | –0.17          |

[1] Nakajima et al. (2023), [2] Nakajima et al. (2022).

- *In Fig. 11b, what is the reason for increased Qfb from not considering cooling towers? This is not because the Qfb,l is now released as Qfb,s, since Qfb = Qfb,l + Qfb,s, right?*

The caption indeed stated "$Q_{FB}$", but the sensible $Q_{FB}$ was in fact plotted. When the cooling tower was considered, the $Q_{FB}$ decreased. We apologise for our error. We have changed the Figure notation from "$Q_{FB}$" to "$Q_{FB\_S}$".

*Technical corrections/comments*

- *L35 - 36: I suggest the authors keep the abstract free of acronyms/jargons and avoid mentioning specifics like "by setting the AHOPTION option in URBPRAM.TBL to 2", which would only make sense to WRF SLUCM users, whereas abstract should be written for a broader audience.*

We have changed the text.
Lines 16–19:
"This method allows users to simulate the dynamic $Q_F$ and the electricity consumption (*EC*) as the outdoor temperature, building insulation, and heating and air conditioning (HAC) performance change. This is achieved via simple selection of certain $Q_F$ options among the urban parameters of WRF."

- *Table 1: CLMU now has the capacity to consider partial AC in the form of AC adoption rate (Li et al., 2024).*

We have changed Table 1 and now cite Li et al. (2024).

**Table 1: Description of urban canopy parameterisations.**

| | SLUCM[1] | **SLUCM+BEM** | BEP+BEM[2] | CM-BEM[3] | CLMU[4,5] | BEM-TEB[6] |
|---|---|---|---|---|---|---|
| $Q_F$ from buildings | Prescribed | **Dynamic** | Dynamic | Dynamic | Dynamic | Dynamic |
| $Q_F$ from traffic | Prescribed | **Prescribed** | – | Prescribed | Prescribed | Prescribed |
| Internal heat gain | – | **Input** | Input | Input | – | Input |
| $EC_{HAC}$ | – | **Dynamic** | Dynamic | Dynamic | Dynamic | Dynamic |
| Partial AC | – | **Implemented** | – | Implemented | Implemented | – |
| COP | – | **Dynamic** | Constant | Dynamic | Constant | Dynamic |
| Cooling tower | – | **Implemented** | – | Implemented | – | – |
| Windows | – | – | Implemented | Implemented | – | Implemented |
| Ventilation | – | – | Implemented | Implemented | Implemented | Implemented |
| Weekday/weekend difference | – | – | – | Implemented | – | – |

AC, air conditioning; BEM, building energy model, BEP, building effect parameterisation; CLMU, community land model–urban; CM, canopy model; COP, coefficient of performance; EC, electricity consumption; $Q_F$, anthropogenic heat; SLUCM, single-layer urban canopy model; TEB, town energy balance.

[1] Kusaka et al. (2001), [2] Salamanca et al. (2010), [3] Kikegawa et al. (2003), [4] Oleson and Feddema (2020), [5] Li et al. (2024), [6] Bueno et al. (2012).

- *L108: missing ";" after "EC, electricity consumption".*

We have revised the text.

- *L140: missing ")" after "Oleson and Feddema 2020".*

We have revised the text.

- *L149: AHOPTION and its options have not been mentioned, which may make it confusing to readers. I suggest either keeping it more general without mentioning the name of this setting, or explicitly refer readers to section 2.1.*

We have revised the text.
Lines 108–110:

"Specifically, we sought to render SLUCM+BEM usable by those who employ both WRF and the original SLUCM. Users simply change certain $Q_F$ options (AHOPTION) in the urban parameter setting file (URBPRAM.TBL) of WRF 1 and 2 (please see Section 2.1)."

- *L171: it seems AHOPTION = 0 represents Qf off. So it should be "… off or on by selecting 0 or 1 …, respectively".*

We have revised the text.

- *Fig. 1: it is overall very difficult to follow due to the number of WRF-specific variables/settings used in this figure. I would suggest defining these variables in the caption, or replace them with actual names (like "wall temperature") or mathematical symbols (something like T_wall) and define the symbols in the caption. Also, it should be made clear (either by mentioning in the caption or labeling it in the figure) that the box with the dashed line is the legend. Keeping only one of those legend boxes (the one in Figure 1c) is sufficient.*

We have revised the text.

[Figure]

[Figure]

**Figure 1: Schematic of energy budgets for an urban canopy layer that includes buildings. The single-layer urban canopy model (SLUCM) with (a) "Zero-Flux" and (b) "Constant" settings. (c): The updated SLUCM based on a building energy model (BEM), thus SLUCM+BEM, with a "Constant" setting. Blue and yellow highlighting**

indicate variables simulated by SLUCM and SLCUM+BEM respectively. The text in the callouts indicates original or newly introduced inputs to the WRF parameter table URBNPRAM.TBL.

- *L213 – 216: it might be better to reword this sentence so that each term is explained separately.*

We have revised the text.
Lines 163–165:
"The first term is the *HTRANS* estimated using Eq. (1) (positive in summer and negative in winter). The second and third terms are the internal sensible heats generated by equipment and the occupants respectively (and are always positive). In the terms,"

- *L270 – 272: where is Qfb,s returned to?*

We have revised the text.
Lines 212–213:
"$Q_{FB\_S}$ and $Q_{FB\_L}$ were respectively added to the sensible and latent heat fluxes, and the results returned to the atmospheric first layers of the meteorological and climate models respectively."

- *Table 2 is not very clear. I suggest formatting this table like the table in Fig. 12, so that it is clear on which rows these two models are sharing parameters. Also explain which set of TRLEND/TBLEND is for summer and which is for winter.*

We have revised the text.

[revised manuscript text omitted]

* Newly added to SLUCM+BEM; (–) dimensionless parameter.

- *L574: seems to be referencing Table 2 instead of Table 1.*

We have revised the text.

- *Fig. 11: for the maps, would it be better to present the differences in each case? E.g., instead of the current panel b map, present the difference between b map and a map (b minus a), and instead of panel c map, present c minus b, etc. This may make the effect more apparent. The diurnal profiles can be kept the same.*

We have revised the Figure and text as follows.

[Figure]

**Figure 11: The average, SLUCM+BEM-simulated $Q_{FB\_S}$ distributions over the Tokyo Metropolitan Area at 14:00 LT in summer (left). Diurnal changes in the $Q_{FB\_S}$ values for Tokyo (BC), Kumagaya (Rm), and Nerima (Rd) (right). Lines and error bars are the simulated average values and the 5th–95th percentiles respectively. The simulations were run for (a) control (CTRL), (b) no cooling tower, (c) no coefficient of performance (COP) change, and (d) no partial HAC scenarios.**

Lines 507–532:
"Figure 11b shows the difference when cooling towers were and were not (No cooling tower - CTRL) considered. As only offices feature cooling towers, the results for residential areas are similar to those obtained previously. When focusing only on offices, the values for central Tokyo were more significant than those shown in Figure 11a. In terms of temporal variation in Tokyo,

the $Q_{FB\_S}$ curve was the same as that described in the previous case, but the peak day value was over 40 W m$^{-2}$, higher than the peak of about 35 W m$^{-2}$ for the control scenario (Fig. 11a). Thus, cooling towers afforded an average day difference of approximately 15 W m$^{-2}$.

Next, we considered the effect of COP changes. Figure 11c shows the difference between a scenario that does not consider COP changes (thus where COP is fixed ["No COP change"]) and a scenario with no cooling tower ("No COP change–No cooling tower). The effects of COP changes were less than those illustrated in Figure 11b. Figure 11c reveals almost no change in the $Q_{FB\_S}$ and that the temporal changes were near-identical at the three representative points. However, $Q_{FB\_S}$ changes should probably be considered when dealing with heat waves and as the urban climate becomes increasingly affected by global warming. The temperatures would then be significantly higher than those of the present study, lowering the COP and increasing the $EC$ and $Q_{FB\_S}$ (Takane et al., 2019; 2020).

Finally, we considered the impact of partial HAC. We changed the settings of Figure 11c to incorporate a whole-of-house HAC (similar to BEP+BEM). We did not consider partial HAC use. Compared to the previous case, the $Q_{FB\_S}$ for the entire metropolitan area increased in the whole-of-house HAC scenario (Fig. 11d). The temporal changes at the three representative locations were also clearly affected. For example, in Tokyo, the nighttime $Q_{FB\_S}$ was greater for the whole-of-house HAC than the partial HAC scenario, and the difference between the daytime and nighttime values smaller. $Q_{FB\_S}$ was approximately 90 W m$^{-2}$ regardless of the time of day. Kumagaya exhibited no significant variation in the diurnal pattern, but the absolute values were consistently above 40 W m$^{-2}$. In Nerima, the pattern shifted to a diurnal peak. Thus, consideration of partial HAC status critically impacted our results. When including partial HAC in a model, new parameters such as those listed in Table 1 are needed to reflect accurately the effects of human activity. These (slightly) complicate the analysis. However, the difference between the No partial HAC and No COP change scenarios (Fig. 11d) illustrates the need to consider partial HAC whenever possible; this strongly impacts the results. Social big data on the population, and electricity and HAC use, will be valuable. Such data were used by Takane et al. (2022) to establish the parameters described above."

- *L660 – 661: I think the focus of this sentence should be how SLUCM+BEM improves upon the "inadequate representation of building energy" in other single-layer UCMs, rather than emphasizing that SLUCM+BEM is the only single-layer UCM with a BEM that is coupled with WRF. This seems to downplay the importance of your work, as your model development efforts, although implemented in WRF, could be adapted and applied to other single-layer UCMs.*

Thank you. We have revised the text:
Lines 573–574:
"The SLUCM improvement that we achieved via implementation of a simple BEM could be extended to other single-layer UCMs."

- *L691 – 692: it is not accurate to say, "heat pumps are positioned as a renewable energy source". Heat pumps by themselves are not a renewable energy source; rather, implementing*

*heat pumps is a way to electrify buildings, which then can make use of renewable energy sources once we decarbonize our grid.*

We have revised the text:
Lines 624–626:
"In addition, parameterisation based on the air-source heat pump AC will become increasingly important in future scenarios. Heat pumps aid decarbonisation and, thus, are attracting increasing attention. Such pumps will become widely used to ensure energy security."

- *L706: "skilfully" should be "skillfully".*

We have revised the text.

**Reviewer #2**

*Summary*

*This well-motivated and well-written study will help to improve urban modeling using the SLUCM and have far-reaching impacts, especially since the SLUCM is the preferred UCM of WRF users. The methods and results are clear. I have some clarifying questions and a few small comments. Otherwise, the manuscript is ready for publication.*

Thank you very much for your review and the positive comments. We have revised the text accordingly.

*Major Comments*

*General: It is claimed in the abstract that the SLUCM-BEM can be applied to climates worldwide but was only simulated over Tokyo. The conclusion mentions future studies can do work of this nature, so right now it cannot be claimed that SLUCM+BEM can be used worldwide. Can an analysis be done for a distinctly different climate than Tokyo, but in a similar fashion? Some of the assumptions made may not be appropriate for different climates compared to Tokyo. How would the model work in more developing cities that may not have all the urban morphology data? The manuscript need not show all the same figures but perhaps a few highlighting similarities and differences. If it is too much for this manuscript, the worldwide claim should be removed from the abstract.*

Thank you. We agree that many more parameters would be required were the model to be applicable worldwide. This is future work. We have deleted the text.

*L19: AHOPTION and URBPRAM.TBL are jargon specific to urban modeling that a general audience will not have knowledge of. Suggest revising to define the terms first before mentioning.*

We have changed the text to read:
Lines 16–19:
"This method allows users to simulate the dynamic $Q_F$ and the electricity consumption (*EC*) as the outdoor temperature, building insulation, and heating and air conditioning (HAC) performance change. This is achieved via simple selection of certain $Q_F$ options among the urban parameters of WRF."

*L41-42: Doesn't a third UCM option exist? The MLUCM/BEP on its own, i.e., not combined with the BEM. Why is that not mentioned?*

In the original manuscript, we wrote:
Lines 48–52:
"WRF employs two main UCM options: the UCM alone, and a combined building energy model (BEM). The UCM alone corresponds to the single-layer UCM (SLUCM, Kusaka et al., 2001;

Kusaka and Kimura, 2004), and a building effect parameterisation (BEP) (Martilli et al., 2002), whereas in the combined building energy model, the BEM is coupled to the BEP to construct BEP+BEM (Salamanca et al., 2010)."

*L95: Describe AHOPTION and URBPRAM.TBL so that those not as familiar with urban modeling understand these terms. Additionally, describe what a change from AHOPTION from 1 to 2 means, i.e., what does 1 mean and what does 2 mean. This is later described in the Methods Section, but readers may be initially confused.*

We have changed the text to read.
Lines 108–110:
"Specifically, we sought to render SLUCM+BEM usable by those who employ both WRF and the original SLUCM. Users simply change certain $Q_F$ options (AHOPTION) in the urban parameter setting file (URBPRAM.TBL) of WRF 1 and 2 (please see Section 2.1)."

*L214-216: Why were the full seasons not simulated, i.e., 01 June to 31 August and 01 December to 28 February? Additionally, why were these years chosen, and not a more recent season?*

We have added the following text:
Lines 246–250:
"In Tokyo, the HAC is generally used only summer and winter seasons (not those of spring and autumn) (Takane et al., 2017). Spring and autumn do not affect the $EC_{HAC}$ and $Q_{FB}$ evaluations simulated by SLUCM+BEM. Thus, no 1-year simulation was performed. The 2018 and 2017 summer and winter were selected because these are the years for which the measurements of $EC$ are available (Nakajima et al., 2022), and there were more clear sky days in these than in other years."

*Section 3.2.1: Why were 05:00 and 14:00 LT analyzed? To capture minimum and maximum temperature?*

Yes. We have added the following text:
Lines 373–376:
"For example, SLUCM+BEM reproduced the observed urban heat island centred on Tokyo well (Fig. 5b) at 05:00 LT (when the temperature was lowest) (Fig. 5a), and observed high temperatures in the inland area at 14:00 LT (when the temperature was highest) (Fig. 5d) were similarly well reproduced (Fig. 5c)."

*General: How does runtime compare between SLUCM and SLUCM+BEM? Can this be added to the manuscript somewhere?*

Thank you. We have added the following text:
Lines 565–568:

"In our simulation environment (HPE Apollo 2000 [scalar computer], 3,072 GFlops, 192 GiB memory, Intel Xeon Gold 6148, 40-core parallel computing, Intel compiler), the computation times for the entire SLUCM+BEM and SLUCM simulations were very similar."

Line 681:
"The computation times for the entire SLUCM+BEM and SLUCM simulations were very similar."

*Minor Comments*

*L12: Remove "worldwide". "Widely used" is sufficient, otherwise redundant.*

We have made the revision.

*Fig 10: Put summer and winter in different colors and add legend for the colors. Initially is confusing that there are two distinct groups for each plot until reading the caption.*

We have made the revision.

[Figure]

**Figure 10: Scatterplots of 2-m temperature and (a) electricity consumption (*EC*), and (b) anthropogenic sensible heat from buildings (*$Q_{FB\_S}$*) in Tokyo (BC), Kumagaya (Rm), and Nerima (Rd) at 14:00 LT in summer and winter simulated by SLUCM+BEM. Each plot shows daily results. Lines with error bars are single regression lines. Plots with temperatures > 20°C represent calculation results for summer; those with temperatures < 20°C represent calculation results for winter.**

*Several figures should have the axes labels and legend text larger. These are Figures. 2, 3, 4, 5, 7, 9, and 11.*

We have made the revision. Please see these figures in the revised manuscript.

*L413: Revise to "…and reaches…".*

We have made the revision.

---

## Author Comment (AC3)

**SLUCM+BEM (v1.0): A simple parameterisation for dynamic anthropogenic heat and electricity consumption in WRF-Urban (v4.3.2)**

Yuya Takane[1], Yukihiro Kikegawa[2], Ko Nakajima[1], Hiroyuki Kusaka[3]

[1]Environmental Management Research Institute, National Institute of Advanced Industrial Science and Technology (AIST), Tsukuba, 305-8569, Japan
[2]School of Science and Engineering, Meisei University, Tokyo, 191-8506, Japan
[3]Center for Computational Sciences, University of Tsukuba, Tsukuba, 305-8577, Japan

*Correspondence to*: Yuya Takane (takane.yuya@aist.go.jp)

**Abstract.** We propose a simple dynamic anthropogenic heat ($Q_F$) parameterisation for the Weather Research and Forecasting (WRF)-single-layer urban canopy model (SLUCM). The SLUCM is a remarkable physically based urban canopy model that is widely used. However, a limitation of SLUCM is that it considers a statistically based diurnal pattern of $Q_F$. Consequently, $Q_F$ is not affected by outdoor temperature changes and the diurnal pattern of $Q_F$ is constant throughout the simulation period. To address these limitations, based on the concept of a building energy model (BEM), which has been officially introduced in WRF, we propose a parameterisation to dynamically and simply simulate $Q_F$ from buildings ($Q_{FB}$) through physically based calculation of the indoor heat load and input parameters for BEM and SLUCM. This method allows users to simulate the dynamic $Q_F$ and the electricity consumption ($EC$) as the outdoor temperature, building insulation, and heating and air conditioning (HAC) performance change This is achieved via simple selection of certain $Q_F$ options among the urban parameters of WRF. SLUCM+BEM was shown to simulate temporal variations of $Q_{FB}$ and EC for HAC ($EC_{HAC}$) and broadly reproduce the $EC_{HAC}$ estimates of more sophisticated BEM and $EC_{HAC}$ observations in the world's largest metropolis, Tokyo.

**1 Introduction**

In the current era of climate change, cities are among the most critical sites for climate change mitigation and adaptation. With urban development, population concentration and urban warming, cities consume more energy and emit more greenhouse gases (GHGs) and anthropogenic waste heat ($Q_F$) than ever. As a result, global and local urban warming will continue to increase (IPCC, 2021; Takane et al., 2019; 2020; Kikegawa et al., 2022). Against this backdrop, climate change mitigation efforts toward the goal of carbon neutrality by 2050 are gaining momentum in countries across development stages, and urban climate change adaptation efforts are also progressing. However, in countries and regions where urban areas are expanding due to population and economic growth, GHG and $Q_F$ emissions associated with urbanisation are expected to continue to increase. In addition, energy consumption, particularly for air conditioning (AC), is predicted to increase under continued global warming in developed and other countries (IEA 2018). Therefore, clarifying the current state of energy consumption,

This method allows model users to simulate dynamic $Q_F$ and electricity consumption ($EC$) according to factors such as outdoor temperature changes, 
[revised manuscript text omitted]

**SLUCM+BEM**

**a MBE**

| | 00 | 01 | 02 | 03 | 04 | 05 | 06 | 07 | 08 | 09 | 10 | 11 | 12 | 13 | 14 | 15 | 16 | 17 | 18 | 19 | 20 | 21 | 22 | 23 | Mean |
|---|---|---|---|---|---|---|---|---|---|---|---|---|---|---|---|---|---|---|---|---|---|---|---|---|---|
| Rm | -0.1 | 0.3 | 0.5 | 0.6 | 0.5 | 0.4 | -0.3 | -1.3 | -1.4 | -1 | -0.4 | 0.2 | 0.7 | 0.8 | 1 | 1 | 0.9 | 0.5 | 0 | -0.1 | -0.2 | -0.4 | -0.5 | -0.5 | 0 |
| Rd | -1 | -0.7 | -0.5 | -0.5 | -0.3 | -0.4 | -1.2 | -2.1 | -2.2 | -2 | -1.6 | -1.2 | -0.9 | -0.7 | -0.6 | -0.6 | -0.8 | -1.3 | -1.8 | -1.8 | -1.8 | -1.9 | -1.7 | -1.5 | -1.2 |
| BC | 0.5 | 0.1 | 0 | 0 | 0.2 | 0.2 | -0.2 | -0.8 | -0.8 | -0.3 | 0.7 | 1.6 | 2.2 | 2.7 | 2.9 | 3 | 2.8 | 2.6 | 2.3 | 2.2 | 2.2 | 1.6 | 1.1 | 0.7 | 1.1 |
| All urban grids | -0.7 | -0.4 | -0.3 | -0.2 | -0.1 | -0.2 | -0.9 | -1.8 | -2 | -1.7 | -1.2 | -0.7 | -0.3 | -0.1 | 0 | 0 | -0.2 | -0.6 | -1.1 | -1.1 | -1.2 | -1.3 | -1.2 | -1.1 | -0.8 |

MBE (W floor – m$^{-2}$)

**b MAE**

| | 00 | 01 | 02 | 03 | 04 | 05 | 06 | 07 | 08 | 09 | 10 | 11 | 12 | 13 | 14 | 15 | 16 | 17 | 18 | 19 | 20 | 21 | 22 | 23 | Mean |
|---|---|---|---|---|---|---|---|---|---|---|---|---|---|---|---|---|---|---|---|---|---|---|---|---|---|
| Rm | 0.9 | 0.9 | 0.9 | 0.9 | 0.8 | 0.7 | 0.8 | 1.4 | 1.6 | 1.3 | 1 | 1 | 1.2 | 1.2 | 1.4 | 1.4 | 1.3 | 1.1 | 1 | 1 | 1.1 | 1.1 | 1.1 | 1.1 | |
| Rd | 1.1 | 0.9 | 0.7 | 0.7 | 0.6 | 0.6 | 1.2 | 2.1 | 2.3 | 2 | 1.6 | 1.3 | 1 | 0.9 | 0.9 | 0.9 | 1 | 1.4 | 1.8 | 1.8 | 1.9 | 1.9 | 1.8 | 1.5 | 1.3 |
| BC | 1.3 | 1.1 | 1 | 1 | 1 | 1 | 1 | 1.4 | 1.6 | 1.7 | 2.2 | 2.7 | 3.1 | 3.3 | 3.4 | 3.5 | 3.3 | 3.2 | 2.9 | 2.8 | 2.7 | 2.2 | 1.8 | 1.6 | 2.1 |
| All urban grids | 1.1 | 0.9 | 0.8 | 0.7 | 0.7 | 0.7 | 1.1 | 1.9 | 2.1 | 1.9 | 1.6 | 1.4 | 1.2 | 1.2 | 1.2 | 1.2 | 1.2 | 1.5 | 1.8 | 1.8 | 1.8 | 1.8 | 1.6 | 1.5 | 1.4 |

MAE (W floor – m$^{-2}$)

**CM-BEM**

**a MBE**

| | 00 | 01 | 02 | 03 | 04 | 05 | 06 | 07 | 08 | 09 | 10 | 11 | 12 | 13 | 14 | 15 | 16 | 17 | 18 | 19 | 20 | 21 | 22 | 23 | Mean |
|---|---|---|---|---|---|---|---|---|---|---|---|---|---|---|---|---|---|---|---|---|---|---|---|---|---|
| Rm | 0.7 | 0.9 | 0.8 | 0.7 | 0.5 | 0.2 | -0.3 | -1.1 | -0.7 | -0.5 | 0 | 0.1 | 0.2 | 0.2 | 0.2 | 0.2 | 0.1 | -0.2 | -0.3 | -0.2 | -0.1 | 0 | 0.3 | 0.3 | 0.1 |
| Rd | 1.3 | 1.4 | 1.3 | 1.1 | 0.9 | 0.4 | 0.8 | 1 | 0.8 | 0.2 | -0.2 | -0.3 | -0.3 | -0.5 | -0.6 | -0.7 | -0.7 | -0.7 | -0.5 | 0.3 | 0.6 | 0.7 | 0.8 | 0.9 | 0.3 |
| BC | -0.4 | -0.3 | -0.3 | -0.3 | -0.2 | -0.2 | -0.4 | -0.6 | 0 | -1.3 | -1.1 | -1.8 | -1.8 | -1.7 | -1.6 | -1.5 | -1.5 | -1.4 | -1.5 | -1.2 | -1 | -0.9 | -0.8 | -0.7 | -0.9 |
| All urban grids | 0.9 | 1.1 | 0.9 | 0.8 | 0.6 | 0.3 | 0.2 | -0.1 | 0 | -0.2 | -0.2 | -0.2 | -0.2 | -0.2 | -0.3 | -0.3 | -0.4 | -0.5 | -0.5 | 0 | 0.2 | 0.2 | 0.4 | 0.6 | 0.1 |

MBE (W floor – m$^{-2}$)

**b MAE**

| | 00 | 01 | 02 | 03 | 04 | 05 | 06 | 07 | 08 | 09 | 10 | 11 | 12 | 13 | 14 | 15 | 16 | 17 | 18 | 19 | 20 | 21 | 22 | 23 | Mean |
|---|---|---|---|---|---|---|---|---|---|---|---|---|---|---|---|---|---|---|---|---|---|---|---|---|---|
| Rm | 1.2 | 1.3 | 1.2 | 1 | 0.9 | 0.7 | 0.8 | 1.3 | 1.2 | 1.1 | 1.1 | 1 | 1 | 1 | 1.1 | 1 | 0.9 | 1 | 1 | 1 | 1.1 | 1.1 | 1 | | |
| Rd | 1.7 | 1.7 | 1.6 | 1.4 | 1.2 | 0.8 | 1.2 | 1.6 | 1.5 | 1.3 | 1.1 | 1 | 0.9 | 1 | 1 | 1 | 1.1 | 1.1 | 1.2 | 1.3 | 1.3 | 1.4 | 1.4 | 1.2 | |
| BC | 1.5 | 1.4 | 1.3 | 1.2 | 1.1 | 1.1 | 1.3 | 1.5 | 1.5 | 2.1 | 2.1 | 2.5 | 2.5 | 2.5 | 2.4 | 2.3 | 2.3 | 2.3 | 2.3 | 2.1 | 2 | 1.9 | 1.7 | 1.7 | 1.9 |
| All urban grids | 1.5 | 1.5 | 1.4 | 1.2 | 1 | 0.8 | 1 | 1.4 | 1.3 | 1.2 | 1.2 | 1.1 | 1.1 | 1.1 | 1.1 | 1.1 | 1.1 | 1.1 | 1.2 | 1.2 | 1.2 | 1.3 | 1.3 | 1.2 | |

MAE (W floor – m$^{-2}$)

[revised manuscript text omitted]